# Optogenetic patterning generates multi-strain biofilms with spatially distributed antibiotic resistance

Xiaofan Jin [1,2] ✉ & Ingmar H. Riedel-Kruse [3] ✉

Spatial organization of microbes in biofilms enables crucial community function such as division of labor. However, quantitative understanding of such emergent community properties remains limited due to a scarcity of tools for patterning heterogeneous biofilms. Here we develop a synthetic optogenetic toolkit 'Multipattern Biofilm Lithography' for rational engineering and orthogonal patterning of multi-strain biofilms, inspired by successive adhesion and phenotypic differentiation in natural biofilms. We apply this toolkit to profile the growth dynamics of heterogeneous biofilm communities, and observe the emergence of spatially modulated commensal relationships due to shared antibiotic protection against the beta-lactam ampicillin. Supported by biophysical modeling, these results yield in-vivo measurements of key parameters, e.g., molecular beta-lactamase production per cell and length scale of antibiotic zone of protection. Our toolbox and associated findings provide quantitative insights into the spatial organization and distributed antibiotic protection within biofilms, with direct implications for future biofilm research and engineering.

Bacterial biofilms are spatially structured communities of surface-adherent microbes[1,2] that have a well-established clinical and scientific relevance, for instance, biofilm bacteria have increased antibiotic tolerance compared to their planktonic counterparts and are associated with chronic infections[3,4]. Spatial biofilm organization modulates the ecological interactions between constituent strains, leading to complex systems-level behavior such as competition and cooperation[1,2,5–9]. These properties make biofilms a promising target for bioengineering applications such as consortia-based chemical synthesis, bioremediation, and energy conversion[5,10–16]. Natural biofilm communities use two main processes to generate spatial heterogeneity, i.e., through sequential bacterial deposition[2] and through phenotypic differentiation[17]. In contrast, we currently lack analogous engineering tools to achieve such spatial patterning in synthetic multi-strain biofilms with similar higher-order complexity as in natural biofilms[18–22]. Such tools would empower research into natural

biofilms such as identifying mechanisms underlying distribution of labor or ecological cooperation in biofilms, as well as enable practical biofilm applications such as smart biomaterials and distributed consortia-based biosynthesis[16,23–27].

Here, we present a toolkit—'Multipattern Biofilm Lithography' (MBL)—that recapitulates the type of processes that natural biofilms use to generate spatial heterogeneity within their communities (Figs. 1, 2). This toolkit extends our previous work on biofilm patterning—'Biofilm Lithography'[19]—using optogenetically expressing adhesive molecules to the bacterial surface, which allows patterns to be controlled at high resolution by photo-masked optical illumination. Related concepts have been developed by applying different adhesive molecules and optogenetic systems to generate mixed biofilm communities using multiple wavelengths of light with optical spatial control around millimeter resolution[18,21,28–30]. MBL produces patterned biofilms (i) by sequential depositing strains, (ii)

[1]Gladstone Institutes, San Francisco, CA, USA. [2]Department of Biomedical Engineering, University of Calgary, Calgary, Canada. [3]Department of Molecular and Cellular Biology (and by courtesy) Applied Mathematics, Biomedical Engineering, and Physics, University of Arizona, Tucson, AZ, USA.
✉ e-mail: xiaofan.jin@ucalgary.ca; ingmar@arizona.edu

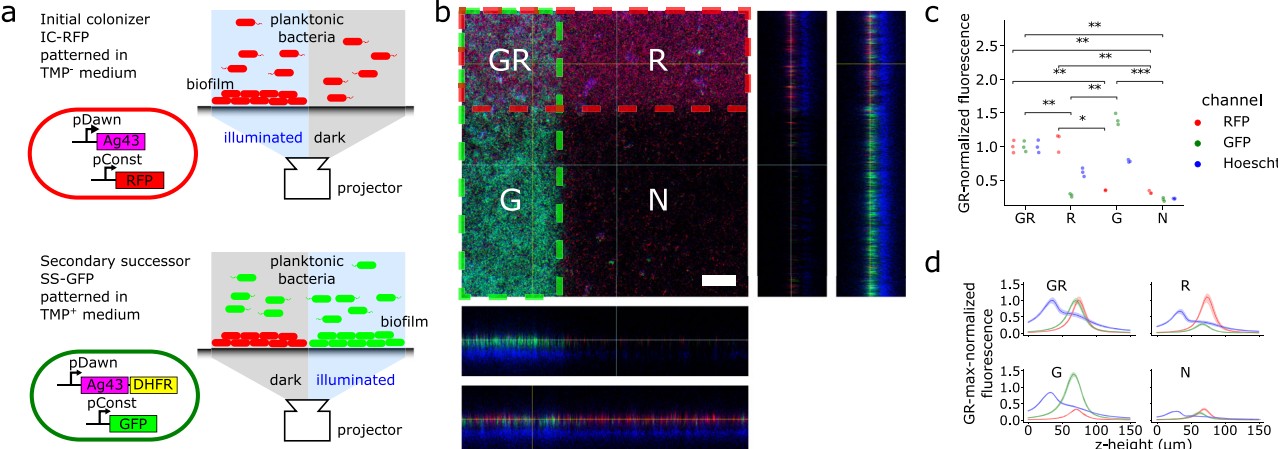

**Fig. 1 | Orthogonal multi-strain biofilm patterning was achieved via sequential optogenetic cell deposition. a** Schematic of sequential patterning protocol: First, an 'initial colonizer' RFP⁺ (IC-RFP) strain carrying optogenetic pDawn-Ag43 was patterned via illumination, then washed out, followed by introduction and illumination of a 'secondary successor' GFP⁺ (SS-GFP) strain carrying pDawn-Ag43+DHFR. DHFR or dihydrofolate reductase confers resistance to trimethoprim and is added to prevent the over-growth of IC-RFP in non-illuminated regions. **b** Confocal image of representative results from (**a**) demonstrated orthogonal patterning of two successive strains in distinct regions. Red and green dashed regions: illuminated regions targeting IC-RFP and SS-GFP strains, respectively; G, R, GR, and N—the expected presence of GFP⁺, RFP⁺, both, and no fluorescence, respectively. XZ and YZ slices are shown below and to the right of the main image, with two slice locations marked in sky blue and yellow, respectively. Scale bar 100 μm. **c** Quantification of fluorescent signal confirmed orthogonality of IC-RFP and SS-

GFP patterning: strong red fluorescence is localized to R and GR regions, strong green fluorescence is localized to G and GR regions, and bacterial biomass measured using Hoechst staining is present in all regions except for the unpatterned N region. Fluorescence is normalized to GR region values, $n = 3$ IC-RFP + SS-GFP co-cultured biofilms. $p$-value annotation for two-sided paired $t$-tests: *$0.01 < p \leq 0.05$, **$0.001 < p \leq 0.01$, ***$p \leq 0.001$. RFP channel $p$-values: $p_{R-vs-G} = 0.011$, $p_{GR-vs-G} = 0.006$, $p_{R-vs-N} = 0.010$, $p_{GR-vs-N} = 0.004$. GFP channel $p$-values: $p_{GR-vs-R} = 0.003$, $p_{R-vs-G} = 0.001$, $p_{G-vs-N} = 0.001$, $p_{GR-vs-N} = 0.005$. **d** Z-height analysis of biofilm taken in the GR, R, G, and N regions indicated that biofilms approach a thickness of 150 μm, with an upper fluorescent layer consisting of GFP⁺ and RFP⁺ cells that rests on top of non-fluorescent biomass (limited oxygen required for GFP and RFP maturation) marked by Hoechst staining. Fluorescence is normalized to maximum GR region values. Data are presented as mean values ± 95% CI, $n = 3$ IC-RFP + SS-GFP co-cultured biofilms.

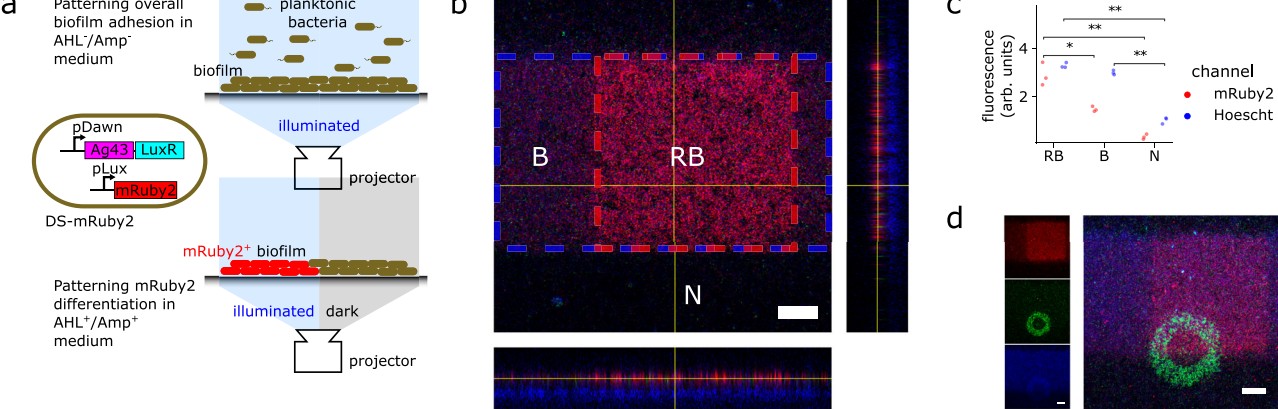

**Fig. 2 | Heterogeneous biofilm patterning was achieved via optogenetic deposition followed by optogenetic differentiation. a** Schematic of differentiation patterning protocol: First, the differentiation strain (DS-mRuby2) was deposited and washed with PBS as before (Fig. 1). Then, culture medium with AHL and ampicillin (growth inhibitor) was added. AHL acts as a secondary chemical switch, allowing light-regulated expression of a target gene (in this case, mRuby2 red fluorescent protein) driven by the pLux promoter. Biofilm regions that were illuminated in this second step produce LuxR. Non-illuminated regions remained non-fluorescent. **b** Confocal image of differentiated patterned biofilm; dashed blue and red regions indicate the illumination zone of the initial patterning step and subsequent differentiation step, respectively. Results indicated that the surface can be independently patterned into distinct regions of biofilm with high mRuby2-red fluorescence (RB), biofilm with low mRuby2-red fluorescence (B), and no biofilm at all (N). Scale bar 100 μm. **c** Quantification of fluorescent signal from (**b**) confirms

orthogonal patterning between initial patterning and subsequent differentiation steps: strongest red fluorescence was localized to (RB) region, while Hoechst signal was consistently present in both (B) and (RB) regions but absent in (N), $n = 3$ DS-mRuby2 differentiated biofilms. $p$-value annotation for two-sided paired $t$-tests: *$0.01 < p \leq 0.05$, **$0.001 < p \leq 0.01$, ***$p \leq 0.001$. RFP channel $p$-values: $p_{RB-vs-B} = 0.022$, $p_{RB-vs-N} = 0.007$. Hoechst channel $p$-values: $p_{B-vs-N} = 0.002$, $p_{RB-vs-N} = 0.001$. **d** The two constructs of this toolkit (Figs. 1a, 2a) are modular and were combined to produce and pattern biofilms with three distinct phenotypes, i.e., generating undifferentiated DS-mRuby2 (low mRuby2-red fluorescence horizontal stripe minus central square), differentiated DS-mRuby2 (high mRuby2-red fluorescence central square), and SS-GFP (GFP⁺ circular ring); RFP, GFP, Hoechst channels shown top, middle, and bottom left respectively, three-channel overlay shown on right, all scale bars 100 μm. Experiment was repeated independently three times with similar results.

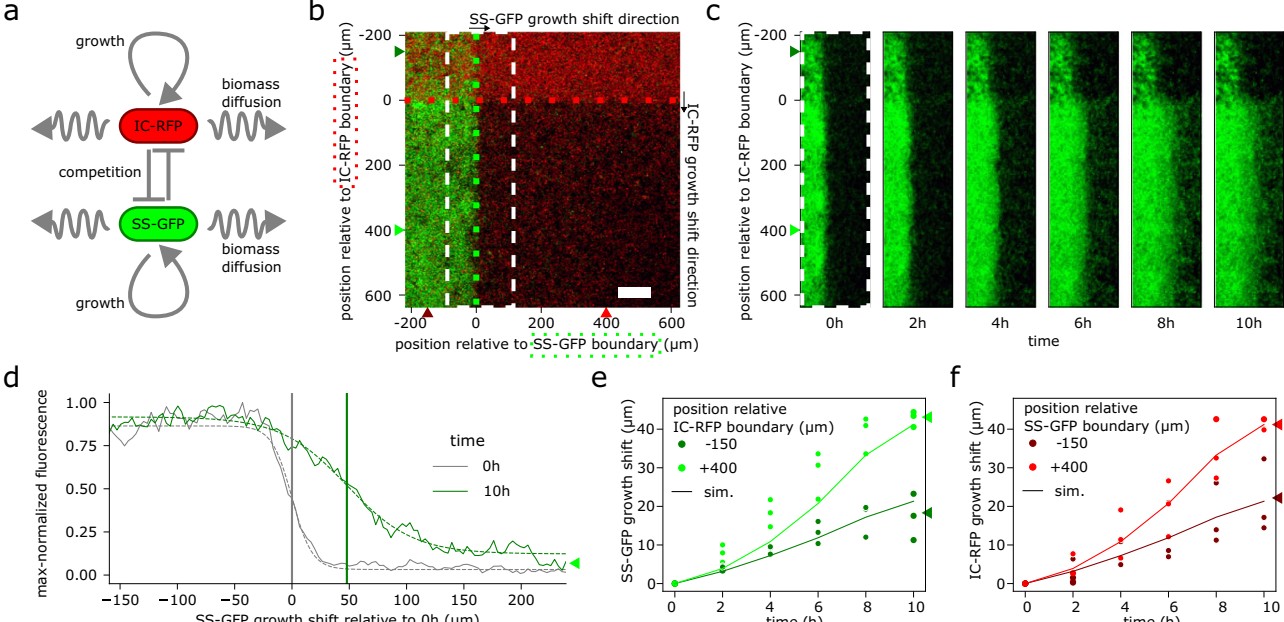

**Fig. 3 | Spatial tracking and biophysical modeling of patterned biofilm enabled quantitative characterization of competitive inter-strain growth dynamics.**
**a** Schematic of a biofilm growth model including red and green biomass representing IC-RFP and SS-GFP strains, respectively, with growth and diffusive expansion modulated by competition for limited nutrient and surface area. **b** Image of initial biofilm with a white dashed box highlighting the region where the growth shift of SS-GFP biofilm (toward the right) is tracked. Vertical axis highlights the position of SS-GFP biofilm relative to the IC-RFP biofilm boundary (dotted red line), with dark and light green triangles, respectively, highlighting −150 and +400 µm positions. Horizontal axis highlights the position of IC-RFP biofilm relative to SS-GFP biofilm boundary (dotted green line), with dark and light red triangles, respectively, highlighting −150 and +400 µm positions. Scale bar 100 µm. Experiment was repeated independently 3 times with similar results. **c** Tracking SS-GFP biofilm growth over time (in the region highlighted by the white dashed box in **b**) reveals outward expansion of biofilm front; expansion was more pronounced outside the IC-RFP biofilm boundary (positions > 0). **d** Automated tracking of moving biofilm front for quantifying biofilm growth based on shifts in the plotted

fluorescence profiles; example shown for position +400 µm from the IC-RFP biofilm boundary, marked by light green triangle in (**b**), for 0 and 10 h time points; dashed curves correspond to logistic function fit (Eq. (1)), gray and green vertical bars indicate extent of the expanding biofilm front inferred from fits at 0 and 10 h, respectively. **e** Experimental quantification SS-GFP biofilm expansion at inner and outer positions (dark and light green points corresponding to −150 and +400 µm relative to the IC-RFP biofilm boundary, respectively) confirmed that growth was slower in the inner position, where direct competition occurred with the opposing IC-RFP strain. Experimental results ($n = 3$ IC-RFP + SS-GFP co-cultured biofilms) were corroborated by numerical predictions of the biophysical model (solid lines). **f** Experimental quantification of IC-RFP biofilm expansion at inner and outer positions (dark and light red points corresponding to −150 and +400 µm relative to the SS-GFP biofilm boundary, respectively) showed that growth is slower in the inner position, where direct competition occurred with the opposing SS-GFP strain. Experimental results ($n = 3$ IC-RFP + SS-GFP co-cultured biofilms) were corroborated by numerical predictions of the biophysical model (solid lines).

by differentiating previously deposited strains, and (iii) by combining (i) and (ii)−all while using just a single wavelength of light. This toolkit is compatible with previously described photo-masking approaches[19] to enable straightforward, orthogonal patterning of heterogeneous biofilm communities at sub-100 µm length scales.

As a demonstration of its practical utility for biofilm research and engineering, we apply this toolkit to explore the nature of spatial microbial interactions during growth and antibiotic exposure (Figs. 3, 4). Earlier work has indicated that resistance to beta-lactams−conferred by expression of beta-lactamase (bla) enzyme−can extend beyond beta-lactamase-producing cells due to diffusion of enzyme and biomass[31–38]. While associated biophysical models have been proposed[39], quantitative characterization has remained challenging due to a lack of experimental platforms. Using MBL, we generate patterned biofilms with spatially heterogeneous beta-lactam resistance and quantify community growth dynamics across space, time, and beta-lactam concentration. In conjunction with computational modeling and a biophysical parameter search, this approach enables a full quantitative characterization of cooperative interactions in our synthetic communities based on distributed beta-lactam protection.

## Results

### Successive bacterial deposition enables orthogonal control of heterogeneous spatially patterned biofilms

To achieve orthogonal patterning of a heterogeneous two-strain biofilm community, we developed an iterative two-round patterning protocol wherein an initial colonizer strain of E. coli MG1655 tagged with mRFP[40] (IC-RFP) and a secondary successor strain tagged with sfGFP[41] (SS-GFP) are arranged independently during successive rounds of patterning (Fig. 1a). In the first step, the IC-RFP strain harboring the pDawn-Ag43 construct was cultured overnight on a well plate using blue-light illumination to generate a patterned single-strain biofilm. Remaining planktonic cells were then rinsed away, and the well plate was inoculated with SS-GFP strain harboring a pDawn-Ag43+DHFR construct. In addition to Ag43 adhesin, the SS-GFP strain was additionally engineered to express dihydrofolate reductase (DHFR) when illuminated, conferring resistance to the growth inhibitor trimethoprim (TMP)[42]. TMP was added to the culture medium for the second patterning step of the SS-GFP strain, during which the second strain was cultured overnight with a new optical illumination pattern, yielding a two-strain biofilm with orthogonal patterning control between the IC-RFP and SS-GFP strains. The addition of TMP, along with the light-regulated expression of DHFR, solved two key challenges for orthogonal patterning: (i) preventing overgrowth of the IC-RFP strain

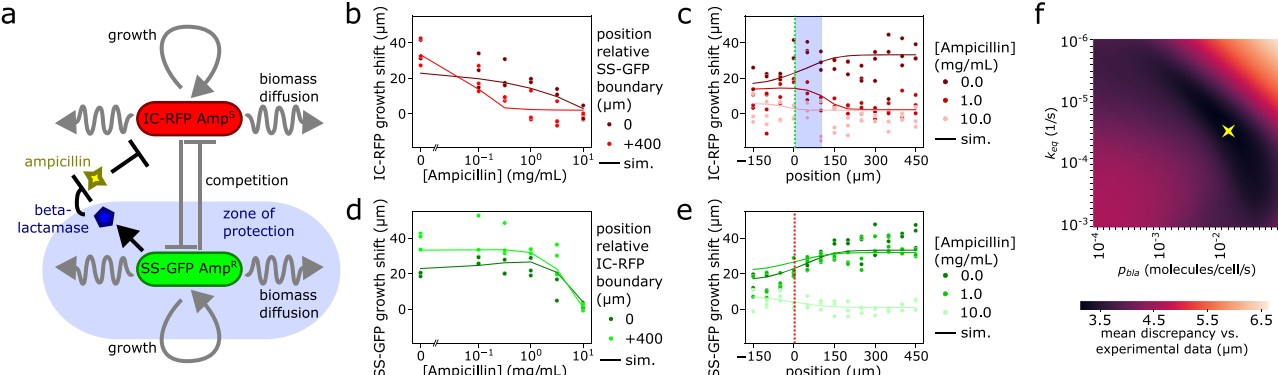

**Fig. 4 | Spatial tracking and biophysical modeling of patterned biofilm under antibiotic exposure revealed conditions for cooperation and a zone of protection. a** Biophysical biofilm growth model (Fig. 3a) was extended with additional inhibition by beta-lactam antibiotics and degradation of these antibiotics by beta-lactamase that was secreted by resistant green biomass representing amp$^R$ SS-GFP biofilm, which led to a zone of protection for otherwise susceptible red biomass (representing Amp$^S$ IC-RFP biofilm). **b–e** Experimental biofilm growth results for the quantification of growth extent across varying antibiotic concentrations for specific spatial positions (**b**, **d**) and along the spatial axis at the 8 h time point for specific antibiotic concentration (**c**, **e**). Experimental results ($n = 3$ IC-RFP + SS-GFP co-cultured biofilms) were corroborated by numerical predictions of the biophysical model (solid lines). **b** Growth of susceptible strain (IC-RFP) decreased with increasing ampicillin antibiotic concentration, and which was noticeably steeper further away from SS-GFP resistant cells (0 vs. 400 µm). **c** Susceptible strain (IC-RFP) grew better when positioned far away from resistant strain (SS-GFP; boundary

indicated with green dotted line) when no ampicillin was present (0 mg/mL); however, at moderate ampicillin concentrations (1 mg/mL), susceptible strain grew faster near resistant cells indicating a zone of protection (yellow shading); no growth of susceptible strain was observed at any position at higher concentrations (10 mg/mL). **d** Growth of resistant cells (SS-GFP) only decreased at high ampicillin concentrations (10 mg/mL); at moderate ampicillin concentrations (0.1–1 mg/mL), growth actually increased compared to the no-ampicillin case due to reduced competition from nearby susceptible cells. **e** Growth of resistant cells (SS-GFP) was consistently greater further away from competing susceptible cells (IC-RFP; boundary indicated with red dotted line as in Fig. 3b). **f** Parameter search using numerical simulations of the antibiotic-extended biophysical model compared against discrepancy with experimental observations generated estimates of media-to-biofilm ampicillin equilibration rate constant ($k_{eq}$) and molecular production rate $p_{bla}$ of beta-lactamase by resistant SS-GFP cells. Optimal values are denoted by a yellow star.

during this step, and (ii) further ensuring that the SS-GFP strain was patterned only in illuminated regions.

To quantify the orthogonality and efficiency of patterning for our protocol, we imaged the patterned biofilms using confocal as well as widefield fluorescent microscopy (Fig. 1b, Supplementary Note 1). Strains were patterned in an intersecting cross pattern with IC-RFP cells in a horizontal stripe and SS-GFP cells in a vertical stripe. We then compared the average pixel fluorescence values in the different surface regions based on illumination conditions. This revealed high levels of red channel fluorescence in the region where only the IC-RFP strain was directed (R) as well as the region where IC-RFP and SS-GFP were both directed (GR). Similarly, we observed high levels of green channel fluorescence in the (G) and (GR) regions, with low fluorescence in both color channels where neither strain was directed (N). We observed greater biomass (i.e., Hoechst staining intensity) in G region compared to R region (Fig. 1d), which we speculated was due to increased the patterning efficiency of SS-GFP strain as growth (including that of planktonic cells) was minimized in unilluminated regions due to presence of TMP. These results confirmed that the successive biofilm patterning protocol is capable of generating orthogonally patterned dual-strain biofilms (Fig. 1c, Supplementary Note 1).

In the vertical Z-dimension, confocal microscopy revealed that the co-cultured biofilms achieved a thickness of approximately 150 µm based on Hoechst staining[43] of biomass (Fig. 1d). The RFP and GFP signals demarcating the IC-RFP and SS-GFP strains respectively were observed primarily in the upper half of the biofilm, consistent with an oxygen gradient preventing fluorescent protein maturation in deeper regions (Fig. 1d). This lack of GFP and RFP signal in anaerobic layers largely prevented the characterization of the degree of vertical mixing or layering between the two strains in the Z-dimension, but also indicated that our approach is capable of engineering biofilms with anaerobic regions, which has many practical applications[16,25]. To quantify the degree of mixing between RFP and GFP cells in the XY-dimensions, we applied spatial cross-correlation analysis between red

and green fluorescent channels, which revealed that populations of each strain were interspersed, with an exclusionary distance on the order of 20 µm (Supplementary Note 2), a length scale consistent with microcolony size in natural biofilms[44]. Collectively, these results confirmed that our successive patterning protocol can generate 2-strain biofilms that are orthogonally patterned with sub-100 µm spatial resolution across a 2D surface, with thickness resembling that of natural biofilms[45].

## Chemo-optogenetic circuit enables spatially regulated biofilm differentiation

In addition to successive colonization, natural biofilms can also develop heterogeneity through spatially regulated differentiation[17]. Inspired by these natural capabilities, we developed an extended optogenetic circuit that enables (i) light-regulated initial patterning as well as (ii) subsequent differentiation of biofilms, applying a two-step illumination process that uses a chemical signal to instruct bacteria to switch between the two steps. This augmented synthetic circuit—pDawn-Ag43+LuxR/pLux-mRuby2—was engineered to have pDawn drive expression of Ag43 as well as expression of the (non-light sensitive) transcriptional regulator LuxR. LuxR drives expression from the pLux promoter in the presence of the chemical signal 3-oxohexanoyl-homoserine lactone (AHL), which drives the expression of the differentiation reporter protein mRuby2[46].

This extended chemo-optogenetic circuit allowed us to realize a two-step adhesion/differentiation patterning protocol in bacteria utilizing two different illumination patterns (Fig. 2a). The differentiation circuit was transformed into *E. coli* MG1655 to generate a differentiation strain (DS-mRuby2). DS-mRuby2 was first cultured overnight to generate a patterned, non-fluorescent biofilm by driving Ag43 expression and surface adhesion in illuminated regions. This initial adhesion step occurred in the absence of AHL, meaning mRuby2 expression remained off, given the requirement for AHL to drive the pLux promoter[47].

Subsequently, the remaining planktonic cells were rinsed away, and a new growth medium containing AHL and ampicillin was introduced to the sample. The role of ampicillin was to prevent biofilm growth during differentiation. During this differentiation step, only a subregion within the illuminated region from the initial adhesion step was chosen to remain illuminated. Within this subregion, expression of luxR via pDawn in conjunction with presence of AHL was designed to drive up-regulation of the fluorescent reporter mRuby2[46] via the pLux promoter, which effectively encoded an AND gate requiring both optical stimulation (to generate LuxR) and AHL (Fig. 2a). To minimize cross-signaling between the initial adhesion illumination pattern and the subsequent differentiation illumination pattern, we used LVA-tagging[48] of the LuxR transcriptional regulator gene to promote rapid protein turnover such that LuxR produced by pDawn activation in the initial patterning step is largely degraded by the differentiation step. Furthermore, TMP was added to the culture medium for the second patterning (i.e., differentiation) step to minimize growth.

The experimental results validated this protocol: we patterned DS-mRuby2 biofilm along a horizontal stripe and then illuminated only a central square within this biofilm stripe for the subsequent mRuby2 differentiation pattern (Fig. 2b). Using confocal microscopy with Hoechst-stained biofilms, we quantified both the total biomass (using Hoechst) as well as RFP fluorescent signals (Fig. 2c) in three distinct regions: the central square where we expected red fluorescent biofilm (RB), the rest of the stripe where we expected non-fluorescent biofilm (B), and finally outside the stripe where no biofilm was patterned (N). Quantification of fluorescent signal indicated significant up-regulation of mRuby2 expression in (RB) region relative to (B) and (N) regions, though leaky expression was evident in the (B) region. Meanwhile Hoechst signal was consistently present in both (R) and (RB) regions, and absent in (N) regions as expected. While we observed significant up-regulation of mRuby2 expression in RB relative to B regions, we did also detect leaky expression of the mRuby2 in B regions, which we speculated was exacerbated by LuxR leftover from the initial patterning step that had not yet been degraded. These results confirmed that initial biofilm patterning and subsequent gene expression differentiation were spatially orthogonal.

## Spatially patterned biofilms via combined sequential deposition and differentiation

The two methods in our toolkit—successive adhesion (Fig. 1a) and differentiation (Fig. 2a)—were designed to be complementary to each other and could thus be combined. We demonstrated this by generating biofilms with two distinct strains, where one was further sub-differentiated into two distinct phenotypes (Fig. 2d). We used DS-mRuby2 as the initial colonizer strain, patterning and differentiating this strain using the two-step differentiation protocol as before. We then introduced SS-GFP strain into the culture, and patterned SS-GFP using a third illumination step. Using confocal microscopy, we confirmed the formation of a biofilm population with three distinct phenotypes (Fig. 2d, Supplementary Note 3): (i) undifferentiated DS-mRuby2 (low red-mRuby2 fluorescence, Hoechst-stain only), (ii) differentiated DS-mRuby2 (high red-mRuby2 fluorescence), and (iii) SS-GFP (green fluorescent). These findings demonstrated the ability of MBL to recapitulate, in a spatially controlled manner, the structure formation processes of differentiation and succession as found in natural biofilms[2,17].

## Spatial growth patterns of synthetic biofilm communities reflect ecological competition

As a practical use case, we combined our synthetic toolkit with biophysical modeling to quantitatively investigate how spatial heterogeneity modulates biofilm development due to bacterial growth, diffusive expansion, and inter-strain ecological interactions (Fig. 3a). Using our successive patterning approach, we first generated 2-strain

biofilms (IC-RFP followed by SS-GFP) in an intersecting pattern in the absence of any antibiotics as described (Fig. 1b), followed by a PBS rinse to remove extraneous cells. Next, we introduced fresh growth medium and subsequently tracked biofilm growth using longitudinal confocal microscopy imaging over a 36 h time course (see the "Methods" section) at 2 h intervals, or alternatively with a 72 h end-point measurement using wide-field fluorescent microscopy (Supplementary Note 1).

We observed a clear outward shift of the biofilm growth front at the boundary between patterned and unpatterned regions for both IC-RFP and SS-GFP biofilms (Fig. 3b, c, Supplementary Note 4). No such expansion was observed when refreshed medium was substituted with a PBS control (Supplementary Note 5), indicating that this phenomenon was indeed due to growth. We quantified this shift using an automated curve fitting procedure (Fig. 3d, see the "Methods" subsection "Longitudinal tracking of biofilm expansion") and confirmed a pronounced biofilm growth front expansion (Fig. 3c) during the first 10 h of the confocal timecourse. Past 10 h, we observed increasing levels of background fluorescence generated by planktonic cells, which interfered with confocal imaging of the underlying biofilm in many samples—thus confocal timepoint analyses were limited to the first 10 h. The 72 h endpoint measurement with wide-field fluorescent microscopy also indicated a clear increase in fluorescence of both IC-RFP and SS-GFP biofilms, providing further evidence of the viability of biofilm communities patterned using our toolkit (Supplementary Note 1).

Comparing growth fronts at different locations, we found that the expansion of SS-GFP and IC-RFP biofilms appeared diminished in the (GR) region where both SS-GFP and IC-RFP strains are co-patterned, which indicated a spatially modulated competition between the two strains (Fig. 3c, Supplementary Note 4). To characterize this spatial effect, we quantitatively tracked the biofilm front shift at two distinct spatial positions along the boundary (Fig. 3c): (i) where the two strains overlap, (inner, −150 μm), and (ii) away from the other strain (outer, +400 μm). This represented a measure of the biofilm growth as a function of proximity to the other strain, i.e., closest in the inner position and farthest in the outer position.

While we observed growth at both positions, larger shifts occurred at the +400 μm position for both strains ($p = 1.4 \times 10^{-3}$ and $2.4 \times 10^{-6}$ for RFP+ and GFP+ strains, respectively, $n = 6$). Shifts at the +400 μm position were ~40 μm within the first 10 h, roughly double that of the −150 μm position (Fig. 3c, Supplementary Note 4), where biofilm was expanding into an unclaimed surface (Fig. 3e, f). As a control, we also observed that biofilm expansion is greater at the boundary of single-strain biofilms than dual strain communities, in particular that of single-strain SS-GFP biofilms (Supplementary Note 6).

To provide a theoretical frame of reference for our experimental observations, we developed a biophysical biofilm growth model (see the "Methods" subsection "Biophysical modeling") that simulated growth in multi-strain biofilms (Fig. 3a) based on a 2D reaction-diffusion grid framework[49]. IC-RFP and SS-GFP strains were abstracted as red and green biomass, respectively. Growth of biomass was modeled using Monod kinetics[50] to reflect shared nutrient limitation common to both strains, with biofilm expansion modeled using an effective biomass 'diffusivity' parameter $D_{bm}$. To account for the surface-adherent biofilm mode of growth, we further constrained growth based on a locally limited surface area carrying capacity $b_{max}$, reflecting competition for the unclaimed surface in biofilms. We then performed numerical simulations of this model using initial conditions that match our spatial patterning profile (i.e., intersecting red and green biomass stripes recapitulating experimental SS-GFP and IC-RFP stripes). All model parameters were pre-determined using biophysically realistic estimates (Supplementary Note 7) with the exception of biomass diffusivity $D_{bm}$. We determined this diffusivity from our

experimental data using parameter search to minimize discrepancy against experimental data. We obtained a value of $D_{bm}$ of 0.149 μm²/s (0.95% CI range 0.132–0.164 μm²/s), roughly 3 orders of magnitude less than the effective diffusivity of motile planktonic bacteria[51] (see the "Methods" subsection "Biophysical modeling", Supplementary Note 8). It should be noted that bacterial diffusivity $D_{bm}$ in our case solely exists as an empirical parameter used to describe observed expansion of the biofilm front, rather than a parameter with a well-described mechanistic underpinning. The ability of our biophysical model to capture behavior observed in experimental data suggests that our synthetic biofilm communities exhibit increased competition due to limited resources (e.g., nutrients, unused surface) in a manner modulated by spatial proximity, demonstrating the utility of MBL in complementing biophysical modeling to produce a quantitative characterization of how spatial structure modulates biofilm ecology.

## Spatial patterning modulates cooperative sharing of antibiotic protection in biofilm communities

As an additional practical use case, we extended our combined biofilm modeling/synthetic patterning approach to investigate how spatial community dynamics are affected by antibiotic exposure (Fig. 4a), a particularly relevant topic for biofilms given their well-known resistance against many antimicrobials[4]. We focus on ampicillin, a beta-lactam antibiotic, due to the known ability of beta-lactamase-producing cells to protect spatially proximal cells[31–38]. We engineered a two-strain community as before but now leveraged the fact that the IC-RFP and SS-GFP strains have different antibiotic resistance markers as a consequence of their associated respective plasmid backbones, such that the SS-GFP strain was resistant to ampicillin amp^R due to constitutive expression of beta-lactamase gene, while the IC-RFP strain was sensitive amp^S. Previous work on beta-lactamase has demonstrated that its protective effect in degrading beta-lactams such as ampicillin can extend beyond the producing cell to otherwise beta-lactam-susceptible neighbors (e.g., satellite colonies with ampicillin selection) in a spatially modulated manner[31–38]. Therefore, by introducing ampicillin into the culture medium, we could effectively tune community inter-strain ecology, and we hypothesized that a spatially modulated cooperative mechanism could emerge by which antibiotic protection was shared in co-culture biofilms due to the resistant cells shielding the colocalized susceptible cells.

To test this hypothesis, we performed experiments where we tracked the growth of patterned co-culture biofilms at various concentrations of ampicillin (Fig. 4a, b). We observed, as expected, that overall growth of the susceptible IC-RFP strain decreased as more ampicillin was added to the culture medium (Fig. 4b). Notably, this inhibitive drop-off was steeper in +400 μm spatial positions where IC-RFP biofilm was far from SS-GFP resistant cells (Fig. 4b), and significantly greater still in single-strain susceptible biofilm where resistant cells were not present (Supplementary Note 6). This was indicative of a spatially modulated cooperative relationship where beta-lactamase protection by SS-GFP cells conferred protection to nearby IC-RFP non-resistant biofilms. At low ampicillin concentrations, both IC-RFP and SS-GFP strains consistently exhibited maximal growth in spatial locations away from their counterpart strain, resembling the competitive dynamics seen during growth without ampicillin (Fig. 4c, e). We observed a striking transition at around 0.3–1 mg/mL ampicillin, where the growth of the susceptible IC-RFP strain was higher at the patterning boundary of the SS-GFP strain (position = 0 μm) relative to the +400 μm position (Fig. 4b, c). We did not observe cooperative effects in control experiments where ampicillin antibiotic was substituted for kanamycin, whose aminoglycoside kinase resistance mechanism cannot be shared between cells[52] (Supplementary Note 9). Overall, these results provided evidence for a cooperative relationship modulated by both antibiotic concentration and spatial proximity

and where IC-RFP susceptible biofilms may derive a benefit from the presence of nearby SS-GFP resistant cells.

Focusing on the response of the SS-GFP-resistant biofilm, we observed that these cells are capable of growing at ampicillin concentration up to 3 mg/mL without significant growth inhibition relative to no-antibiotic condition (Fig. 4d)–clear inhibitive effects are observed at 10 mg/mL ampicillin (Fig. 4d, e). Notably, for the SS-GFP biofilm adjacent to IC-RFP biofilms (boundary position), the addition of a low 0.1 mg/mL concentration of ampicillin actually improved growth relative to no ampicillin condition, suggesting that in this context, SS-GFP cells took advantage of reduced competition with IC-RFP cells which were inhibited by ampicillin. Similarly, we observed with endpoint widefield fluorescence measurements of RFP and GFP levels in the GR biofilm region (both strains present) that the addition of low-to-moderate ampicillin concentrations (e.g., 0.1–3 mg/mL) actually increased SS-GFP signal (Supplementary Note 1). These results indicated that by inhibiting the growth of susceptible IC-RFP cells, the addition of increasing ampicillin could remarkably also serve to *improve* the growth of resistant SS-GFP biofilm by reducing its competitors. Altogether our findings pointed to an interplay of both cooperative (SS-GFP cells protected nearby IC-RFP cells) and competitive (limited nutrients and surface area) dynamics in our two-strain biofilms, modulated by both environmental factors (e.g., antibiotic concentration/type) and spatial structure.

To provide a unified framework for understanding these experimental observations, we extended our biophysical model (Fig. 4a, see the "Methods" subsection "Biophysical modeling", Supplementary Note 7). The extended model still used abstract red and green biomass terms to represent IC-RFP and SS-GFP strains from our experiments and was augmented to include terms describing the concentration of ampicillin, and the biofilm concentration of beta-lactamase, which was produced by the green strain at a production rate $p_{bla}$ to reflect the beta-lactamase production in amp^R SS-GFP cells. For simplicity, we assumed that the spread of beta-lactamase was driven by effective biomass diffusion ($D_{bm}$) of the SS-GFP strain. Beta-lactamase was modeled to promote the degradation of ampicillin in the biofilm using first-order enzyme kinetics[53], with ampicillin equilibrating between the biofilm and surrounding culture medium at a rate $k_{eq}$, which defined the timescale required for ampicillin to penetrate into the biofilm. Different inherent ampicillin susceptibilities of the red and green biomass were modeled using Hill dynamics[54], with growth rates for the green biomass modulated by a higher half-max Hill constant to reflect greater inherent resistance (see the "Methods" subsection "Biophysical modeling"). Moreover, the model accounted for the fact that green biomass produced beta-lactamase, which then degraded ampicillin in the biofilm and ultimately lowered the ampicillin concentration in the culture medium.

Model parameters were obtained from biophysically realistic measurements (Supplementary Note 7) with the exception of beta-lactamase production rate $p_{bla}$ and culture medium-to-biofilm equilibration rate of ampicillin $k_{eq}$. We were unable to identify estimates of these two parameters from existing literature, and instead left them as free parameters and estimated their values using a parameter search that minimized mean discrepancy to experimental data (Fig. 4f, see the "Methods" section, Supplementary Note 8). This yielded an estimate for $p_{bla}$ of 0.013 molecules/cell/s (0.95% CI range 0.008–0.014 molecules/cell/s) and an estimate of equilibration rate $k_{eq}$ of $2.86 \times 10^{-5}$ s⁻¹ (0.95% CI range $1.72–3.12 \times 10^{-5}$ s⁻¹), corresponding to a time constant on the order of 10 h for our synthetic biofilm system. This estimate is within a reasonable range of earlier measurements made in biofilms in different species and contexts: previous work estimated the penetration of ampicillin and ciprofloxacin in *Klebsiella pneumoniae* biofilms using the membrane filter disk approach[55], measuring 50% penetration at 40 and 120 min for ampicillin and ciprofloxacin respectively, corresponding to penetration time constants of 0.96 and 2.98 h, respectively assuming an

exponential model. Consistent with the experimental results, the simulated biofilms also exhibited distributed antibiotic protection, with enhanced growth of susceptible red biomass when growing near resistant green biomass at 1 mg/mL ampicillin. This unification of community responses—across space, time and beta-lactam concentrations—in one in-silico model represents a full quantitative characterization of ecological relationships within the biofilm community, demonstrating the utility of our combined modeling and experimental observation approach enabled by MBL.

We also ran experimental controls where the biofilm community was subjected to kanamycin, with accompanying numerical simulations (Supplementary Note 9). Unlike with beta-lactamase, we expected from the literature that protection against kanamycin would not be shared from the resistant strain to the susceptible strain given that the mechanism of resistance (due to aminoglycoside hydrolase) works fully intracellularly[56]. In these experiments and simulations, we found that competitive effects were still observed between the two strains, but no (measurable) zone of protection was found, in agreement with expectations (Supplementary Note 9).

## Discussion

We present 'Multipattern Biofilm Lithography' (MBL), a toolkit for heterogeneous spatial patterning of synthetic biofilms via optogenetics (Figs. 1, 2). This toolkit consists of two modules that recapitulated two key natural mechanisms of biofilm patterning: first, bacteria that colonized a surface were subsequently colonized by other strains or species in an orchestrated manner, resulting in a spatially regulated co-distribution of distinct bacterial strains[2]. Second, bacteria in already established biofilms differentiated into multiple distinct bacterial subphenotypes, which allowed biofilms to optimize cellular functions according to spatial location, such as having outer cells adopt a protective barrier-like phenotype and thereby guard more vulnerable cells in the biofilm interior[17]. In both modules, regulation with high spatial resolution was achieved using optogenetic constructs that were controlled by a single wavelength of light and a photomask.

Our toolkit enabled direct testing of biophysical models describing biofilm spatial ecology. We quantitatively characterized how antibiotic exposure affects the growth dynamics of heterogeneous dual-strain biofilms with differing resistance phenotypes to beta-lactam antibiotics (Figs. 3, 4). Here we identified a shift in the ecological relationship between antibiotic-resistant and susceptible cells in such biofilms when the beta-lactam ampicillin was added to the growth medium: at low or zero levels of ampicillin, competitive effects dominated as the strains compete for limited nutrient and surface area resources, with ampicillin providing a growth advantage to the resistant strain due to inhibition of its susceptible competitor. However, at an intermediate ampicillin concentration, the community ecology shifted towards cooperation, with the resistant strain providing protection against antibiotics for the susceptible cells that were in spatial proximity.

These experimental observations are in quantitative agreement with a biophysical model we constructed from first principles, using a reaction–diffusion framework to describe the growth of biofilm across the surface (Fig. 4). Together with experimental findings from MBL, this model generated a unified quantitative characterization of growth and distribution of antibiotic protection in a heterogeneously resistant biofilm community across space, time, and ampicillin concentration. By comparing numerical simulations with experimental data using parameter space search, we were furthermore able to produce measurements of previously undefined biophysical parameters, including the beta-lactamase production rate $p_{bla}$ of 0.013 molecules/cell/s (0.95% CI range 0.008–0.014 molecules/cell/s) (Supplementary Note 8). This approach also estimated an effective biomass diffusion rate of 0.149 $\mu m^2$/s (0.95% CI range 0.132–0.164 $\mu m^2$/s), and the media-to-biofilm equilibration rate of ampicillin of $2.86 \times 10^{-5}\,s^{-1}$ (0.95% CI

range $1.72–3.12 \times 10^{-5}\,s^{-1}$) corresponding to ~10 h for ampicillin to effectively penetrate biofilm, which aligns reasonably with previous estimates using membrane filter disk approach (albeit in a different bacterial species and context)[55]. We re-emphasize that these quantitative estimates were derived using multi-variable parameter optimization of biophysical modeling to observed data rather than direct experimental measurement, and while they are biologically plausible, nevertheless should be interpreted with a dose of caution. These numbers corresponded to an estimate of the length scale of shared beta-lactam antibiotic protection on the order of 70 $\mu m$ based on the diffusive relationship $D \sim L^2/t$, which aligned with experimental observations (Fig. 4). We note that this estimate is specific to our synthetic biofilm system and would vary in other systems depending for instance on differences in the rate of antibiotic penetration. Overall, our work complemented recent findings by other groups that identified similar zones of protection in microbial communities consisting of resistant and susceptible strains, including protection of gut pathogens[33,36,57] by now introducing a quantitative basis to this effect.

Our toolkit then significantly aids the engineering of key features relevant to natural biofilms and consequently to various applications[58–60]. Specifically, the synthetic circuits control the development and growth of realistic multi-strain biofilms, going beyond the well-established control of exponential/planktonic growth[1]. This then also leads to realistic biofilm conditions inside the synthetic community, such as oxygen gradients, competition, and antibiotic resistance sharing.

Our testing of the genetic constructs has been limited to *E. coli*. Future work in different bacterial species would enable multispecies biofilms with more phylogenetically distinct strains[1,61], and also extend to more complex communities beyond the 2–3 strains demonstrated here. This can be realized by combining with existing techniques for concurrently multiplexing biofilms using multiple light wavelengths[21], thereby taking advantage of both concurrent and sequential adhesion/differentiation.

Concurrent patterning approaches may also enable communities with more similar strain physiology. Our current successive biofilm patterning protocol means that IC-GFP bacteria were patterned on a naive surface and stayed on the surface one day longer compared to SS-GFP bacteria which were patterned on a pre-colonized surface. Patterning of SS-GFP also occurred in the presence of TMP, with pDawn in SS-GFP additionally driving expression of DHFR. This increased the patterning efficiency of SS-GFP strain as growth (including that of planktonic cells) was minimized in unilluminated regions, funneling biomass to the illuminated regions—this was reflected in greater Hoechst staining intensity in G compared to R regions of the biofilm, as well as the observation that single-strain SS-GFP biofilms (where SS-GFP was patterned onto naive polystyrene) exhibited much more rapid growth than their single-strain IC-RFP biofilms (Supplementary Note 6). In sequential co-cultured biofilms, this increase was likely offset by the fact that SS-GFP bacteria were patterned on a pre-colonized (rather than naive) surface, resulting in similar observed growth between IC-RFP and SS-GFP biofilms (Fig. 3e, f).

More complex future systems may also combine optical signaling with chemical signaling, for instance, using quorum sensing genetic modules that generate AHL to control our differentiation circuit (instead of AHL supplied exogenously as in this work), thus incorporating certain members of the population that are able to natively synthesize AHL, and thus act as differentiation signal generators[62,63]. The use of oxygen-independent fluorescent reporters[64] in place of RFP and GFP could also better illuminate spatial structure in anaerobic biofilm regions, a limitation of our current design, which prevented the characterization of spatial strain distribution in the Z-dimension.

On the computational side, more sophisticated modeling approaches, including agent-based biofilm models, will enhance

predictive understanding of more complex ecological interactions such as metabolic cross-feeding, contact-dependent inhibition, division of labor, and antibiotic stress response[65,66], as well as capture finer details of mechanism that have been abstracted away in our current simplified model. For instance, agent-based models could be used to capture biofilm spread more faithfully than our effective biomass diffusivity approach, incorporating distinct terms to account for passive and active motility (e.g., shoving vs. swarming)[65,67]. Additionally, while we have modeled beta-lactam antibiotic effects as a reduction in growth rate (thus preserving model simplicity with death rate modeled by constant first-order decay), in reality, the inhibitory effect of beta-lactams is known to be mediated via growth-rate dependent increase in death rate[68,69]—a mechanism which would require additional terms in the model to capture. Future models can also better reflect subtle differences between constituent strains—for instance, apart from antibiotic resistance, our current model simplistically considers IC-RFP and SS-GFP strains as identical. However, in our current sequential patterning approach, there were additional physiological differences between the IC-RFP and SS-GFP strains in the final patterned biofilms that resulted from their different patterning order and protocols, such as initial density differences between IC-RFP and SS-GFP biofilms across G, R, and GR regions that were not captured in the current simplified biophysical model.

Finally, studying synthetic biofilm systems under flow conditions in addition to static cultures is also of high relevance, as many natural biofilms experience shear stress due to flow that shapes their spatial structure[70]. These advances will ultimately lead to an improved understanding of the structure-function relationship in microbial biofilms, paving the way to platforms that rationally engineer structured, productive, and beneficial microbial consortia using synthetic biology[10].

## Methods

### Plasmids and bacterial strains

All experiments were performed using *E. coli* MG1655 transformed with pDawn-Ag43 plasmid[19] as a base chassis. For successive adhesion, MG1655+pDawn-Ag43 was additionally transformed with a plasmid encoding mRFP expression from a pLac promoter (Biobrick Bba_J04450_pSB4K5 from the iGem parts registry[71]) to generate the IC-RFP strain. Instead of Bba_J04450_pSB4K5, the SS-GFP strain was additionally transformed with a plasmid (pSuccessor) encoding superfolder-GFP (sfGFP)[41] expression from a constitutive promoter along with DHFR expression from a $\lambda$ promoter, and the $\lambda$ promoter is in turn regulated by blue light via the pDawn machinery[72] present on the pDawn-Ag43 plasmid. This pSuccessor plasmid also encodes for ampicillin resistance via the contained beta-lactamase gene. For adhesion followed by differentiation, MG1655+pDawn-Ag43 was additionally transformed with a plasmid (pDifferentiation) encoding mRuby2 driven by the lux promoter, as well as LuxR expression from a pLambda promoter, the pLambda promoter is in turn regulated by blue light via the pDawn machinery[72] present on the pDawn-Ag43 plasmid. Table 1 lists all plasmids used in this work.

### Protocol for successive biofilm patterning

IC-RFP was cultured to late log phase in LB broth under dark conditions (OD600 1.4, ~6 h with shaking at 37 °C after 1:1000 dilution of overnight culture). Culture medium was supplemented with antibiotics as appropriate (50 μg/mL for kanamycin and spectinomycin). This culture was then seeded onto non-tissue-culture treated polystyrene well plates at 1:100 dilution into M63 culture medium[73] supplemented with 0.2%w/v glucose and 0.1%w/v casamino acids (henceforth referred to as M63). Well, plates containing the biofilm cultures were taped to the ceiling of a 37 °C incubator, with a film photomask taped directly under the well plate defining the illumination pattern. An Ivation Pro4 Wireless Pocket Projector (IVPJPRO4) was secured below the ceiling of the incubator, pointing upwards toward the well plate on the incubator ceiling[19,74]. The projector was connected via HDMI cable to a laptop through the side access port of the incubator, and Microsoft PowerPoint software was used to project blue light upward at the well plate with an intensity of 50 μW/cm², modulated using an adjustable neutral density filter (*K&F* concept AMSKU0124) at the aperture of the projector and verified using a Newport optical power meter with UV–vis photodetector (Newport 840C/818-UV). Cultures were then allowed to grow overnight in this initial patterning phase. Culture medium was subsequently aspirated, and wells were gently washed twice with PBS.

During this initial patterning phase, SS-GFP was cultured to late log phase in LB broth under dark conditions (OD600 1.4, ~6 h with shaking at 37 °C after 1:1000 dilution of overnight culture). Culture medium was supplemented with antibiotics as appropriate (50 μg/mL for spectinomycin and 100 μg/mL for ampicillin). This SS-GFP strain culture was then seeded onto the culture plate containing IC-RFP biofilm (after aspirating PBS rinse) at 1:100 dilution into M63 culture medium supplemented with 0.2%w/v glucose, 0.1%w/v casamino acids and 10 μg/mL trimethoprim. Well plates containing the biofilm cultures were illuminated and cultured overnight as described above, using an orthogonal pattern to the IC-RFP biofilm pattern. The culture medium was subsequently aspirated, and wells were gently washed twice with PBS. Multipatterned biofilms were then ready for downstream imaging and/or culturing. Long-term endpoint characterization of biofilm growth was carried out by leaving patterned biofilms in M63 culture medium at room temperature for 3 days, followed by PBS rinse. The sample was then imaged under a Leica DMI6000B wide-field fluorescence microscope using LAS AF software, with separate channels for RFP and GFP.

### Protocol for biofilm patterning followed by differentiation

*E. coli* strain containing pDawn-Ag43 and pDifferentiation (DS-mRuby2 strain) was cultured to late log phase in LB broth under dark conditions (OD600 1.4, ~6 h with shaking at 37 °C after 1:1000 dilution of overnight culture). Culture medium was supplemented with antibiotics as appropriate (50 μg/mL for kanamycin and spectinomycin). This culture was then seeded onto non-tissue-culture treated polystyrene well plates at 1:100 dilution into the M63 culture medium. Well plates containing the biofilm cultures illuminated and culture overnight as described above. Cultures were then allowed to grow

**Table 1 | Plasmids used in this work (plasmids newly generated in this work can be found at https://www.addgene.org/Ingmar_Riedel-Kruse/)**

| Plasmid name | Genes | Antibiotic resistance marker | Origin of replication | Source |
|---|---|---|---|---|
| pDawn-Ag43 | Ag43 regulated by pDawn transcriptional regulator | specR | colE1 | 19 |
| Bba_J04450 (pSB4K5 backbone) | mRFP driven by lac promoter | kanR | pSC101 | 71 |
| pSuccessor | sfGFP driven by constitutive promoter J23100 and DHFR driven by $\lambda$ promoter | ampR | p15A | This work (https://www.addgene.org/228391/) |
| pDifferentiation | mRuby2 driven by pLux promoter and LuxR driven by $\lambda$ promoter | kanR | p15A | This work (https://www.addgene.org/228393/) |

overnight in this initial patterning phase. Culture medium was subsequently aspirated, and wells were gently washed twice with PBS, followed by the introduction of M63 culture medium supplemented with 0.2%w/v glucose, 0.1%w/v casamino acid, 1 μM 3-oxohexanoyl-homoserine lactone (AHL) and 10 μg/mL trimethoprim. Well plates containing the biofilm cultures were illuminated and cultured overnight again using an illumination pattern subset of the initial adhesion pattern. Culture medium was subsequently aspirated, and wells were gently washed twice with PBS. Differentiated biofilms were then ready for downstream imaging and/or culturing.

For heterogenous biofilms generated through a combination of differentiation and successive adhesion, the differentiation protocol described above was followed by addition of SS-GFP strain. This SS-GFP strain culture was then seeded onto the culture plate containing the differentiated biofilm at 1:100 dilution into M63 culture medium supplemented with 0.2%w/v glucose, 0.1%w/v casamino acids and 10 μg/mL trimethoprim. Well plates containing the biofilm cultures were illuminated and cultured overnight as described above, using an orthogonal pattern to the differentiated strain.

## Longitudinal imaging of biofilm expansion

In addition to endpoint imaging of biofilm growth, we also used longitudinal confocal imaging to track biofilm growth of samples prepared using successive adhesion. Biofilms were cultured in M63 culture medium supplemented with 500 ng/mL Hoechst 34580 stain in 12-well plates, and imaged every 2 h for a 36 h timecourse using Zeiss LSM 880 Confocal Microscope using ZEN Microscopy Software, with a ×10 objective, in a heated (30 °C) microscopy chamber. Hoechst, GFP, and RFP signals were captured using excitation/emission wavelengths of 405/455, 488/530, and 561/626 nm, respectively. The imaging focused on an 850 by 850 μm region of interest where a vertical SS-GFP biofilm stripe intersected a horizontal IC-RFP biofilm stripe. Using one well per condition, replicates were obtained by imaging 3 distinct regions in each well/condition. 3D confocal drift correction was applied using FIJI's Correct 3D Drift plugin[75]. Over the course of the time-lapse imaging, background fluorescence from newly grown planktonic cells increased, which eventually made this analysis method unreliable for some samples past the 10 h mark. Accordingly, we focused on the first 10 h of the timecourse in the analysis and used the extent to which the biofilm boundary has expanded after 8h of culture as a consistent benchmark measure of biofilm growth activity. At each given timepoint, for multiple positions along both the GFP and RFP biofilm boundaries, we used a logistic function to fit the fluorescent signal drop-off across the patterned-unpatterned boundary (Fig. 3d) and thus quantitatively characterized the extent of biofilm growth $x0$ over space and time in an automated manner. Fluorescent signal drop-off curve at a given position was generated by averaging a confocal fluorescent signal summed across $Z$-stacks, along a slice with a width of 80 μm centered around the position of interest. Curve fitting was implemented using scipy.optimize.curve_fit[76].

$$\text{Fluorescence} = \frac{A}{(1 + e^{(x-x0)/B} + C)} \quad (1)$$

This analysis was repeated with biofilms cultured at various concentrations of ampicillin, i.e., 0, 0.1, 0.32, 1, 3.16, and 10 mg/mL ampicillin, with biofilm left in PBS as a control. Replicates were generated by imaging three independent stripe intersections per sample.

## M5: Biophysical modeling

To develop a deeper understanding of the relationship between shared antibiotic protection mechanisms, we implemented an in silico biofilm growth model capable of simulating the growth of spatially patterned multi-strain biofilms (Fig. 4a). The model was adapted from a 2D reaction-diffusion grid framework[49] and contains the following 2D state variables:

- $b_G$ is the 2D biomass density of green bacteria representing SS-GFP biofilm producing beta-lactamase, resistant to ampicillin (dimension $M/L^2$)
- $b_R$ is the 2D biomass density of red bacteria representing IC-RFP biofilm, susceptible to ampicillin (dimension $M/L^2$)
- $c_{blaMedia}$ is the local molar concentration of ampicillin in the culture medium (dimension $N/L^3$)
- $c_{blaBiofilm}$ is the local molar concentration of ampicillin in the biofilm (dimension $N/L^3$)
- $c_{bla}$ is the local molar concentration of beta-lactamase enzyme in the biofilm (dimension $N/L^3$)
- $\rho_{nut}$ is the local mass concentration of nutrients (dimension $M/L^3$)

Growth of biomass was modeled using Monod kinetics based on nutrient concentration, parameterized by half-max nutrient concentration $K_{nut}$ (dimension $M/L^3$), and a maximal growth rate $\mu_{max}$ (dimension $1/T$). Growth rates $\mu_R$ and $\mu_G$ (dimension $1/T$) are subject to a locally limited carrying capacity $b_{max}$ (dimension $M/L^2$) to reflect the additional competition for biofilm growth surface. Inhibitory effects of ampicillin in the biofilm were modeled using a Hill term parameterized by cooperativity constant $n$ (dimensionless) and with Hill half-max constants $K_{ampG}$ and $K_{ampR}$ (dimension $M/L^2$) for growth of G and R strains, respectively. We set $K_{ampG} > K_{ampR}$ to reflect greater ampicillin resistance in G strain relative to the susceptible R strain .

$$\mu_R = \mu_{max}\left(\frac{\rho_{nut}}{K_{nut}+\rho_{nut}}\right)\left(1-\frac{b_R+b_G}{b_{max}}\right)\left(\frac{K_{ampR}^n}{K_{ampR}^n+c_{blaBiofilm}^n}\right),$$
$$\mu_G = \mu_{max}\left(\frac{\rho_{nut}}{K_{nut}+\rho_{nut}}\right)\left(1-\frac{b_R+b_G}{b_{max}}\right)\left(\frac{K_{ampG}^n}{K_{ampG}^n+c_{blaBiofilm}^n}\right) \quad (2)$$

Change in biomass over the 2D spatial field was modeled to occur due to growth, death was modeled by first-order decay constant $\gamma_{bm}$ (dimension $1/T$), and outward spread of biomass was modeled with effective diffusivity term $D_{bm}$ (dimension $L^2/T$).

$$\frac{\partial b_R}{\partial t} = (\mu_R - \gamma_{bm})b_R + D_{bm}\nabla^2 b_R,$$
$$\frac{\partial b_G}{\partial t} = (\mu_G - \gamma_{bm})b_G + D_{bm}\nabla^2 b_G, \quad (3)$$

Ampicillin was modeled to equilibrate between the culture medium and biofilm with rate constant $k_{eq}$ (dimension $1/T$). Ampicillin in the biofilm was modeled to degrade enzymatically based on beta-lactamase concentration according to Michaelis–Menten kinetics with catalysis constant $k_{catBla}$ (dimension $1/T$) and dissociation constant $K_{dBla}$ (dimension $M/L^3$), as well as natural self-degradation with first-order exponential decay constant $\gamma_{amp}$ (dimension $1/T$).

$$\frac{\partial c_{blaBiofilm}}{\partial t} = k_{eq}(c_{blaMedia} - c_{blaBiofilm})$$
$$+ \left(\frac{k_{catBla}c_{bla}c_{blaBiofilm}}{K_{dBla}+c_{blaBiofilm}}\right) - \gamma_{amp}c_{blaBiofilm},$$
$$\frac{\partial c_{blaMedia}}{\partial t} = -k_{eq}(c_{blaMedia} - c_{blaBiofilm})$$
$$- \gamma_{amp}c_{blaMedia} + D_{amp}\nabla^2 c_{blaMedia}, \quad (4)$$

We modeled the production of beta-lactamase molecules by resistant green biomass (representing SS-GFP biofilm) using a constant rate $p_{bla}$ (dimension molecule/cell/$T$). Our model includes a $m_{cell}$

parameter (dimension $M$/cell) estimating dry biomass per bacterial cell, Avogadro's constant $N_A$ (dimension molecule/$N$), and height of the culture chamber $h$ (dimension $L$) to translate the effect of beta-lactamase molecular production per cell to changes in overall molar concentration of beta-lactamase as a function of cell biomass density $b_G$ (dimension $M/L^2$). Beta-lactamase is modeled with first-order exponential decay parameterized by constant $\gamma_{bla}$ (dimension $1/T$). Nutrient concentration decreases at a rate proportional to biofilm growth, with biomass yield constant $Y$ (dimensionless)−again, we used the height of the culture chamber $h$ (dimension $L$) to translate between changes in cell biomass density $b_G$ (dimension $M/L^2$) and nutrient mass concentration $\rho_{nut}$ (dimension $M/L^3$). Ampicillin and nutrients in the culture medium were modeled to diffuse with diffusivity terms $D_{amp}$ and $D_{nut}$ (dimension $L^2/T$), respectively.

$$
\begin{aligned}
\frac{\partial c_{bla}}{\partial t} &= \frac{p_{bla}b_G}{N_A m_{cell} h} - \gamma_{bla}c_{bla}, \\
\frac{\partial \rho_{nut}}{\partial t} &= -\left(\frac{\mu_R b_R + \mu_G b_G}{Yh}\right) + D_{bm}\nabla^2\rho_{nut},
\end{aligned}
\tag{5}
$$

Using a forward time-centered space numerical approach, we performed a numerical simulation of this model, using biophysically realistic parameter values and initial conditions−drawn from literature when available (Supplementary Note 7). Simulations assumed the separation of time scales between the diffusivity of ampicillin/nutrients and diffusivity of biomass $D_{amp}$, $D_{nut} \gg D_{bm}$. Green and red biofilm growth extents were calculated by fitting the logistic function (Eq. (1)) for different times, spatial positions relative to stripe boundary, and antibiotic concentrations to enable direct comparison with values obtained experimentally. We performed a 1-D raster search across the parameter values for $D_{bm}$ (effective biofilm biomass diffusivity) to minimize the mean discrepancy between simulated growth extents and those observed in experimental observations with no added ampicillin. To generate an estimated range for the fitted parameter value of $D_{bm}$, we applied a bootstrap approach where we generated 1000 sets 'simulated' experimental observations by drawing growth extents from a random normal distribution parameterized by the observed mean and standard deviation at each timepoint and position. We then generated 95% confidence intervals by taking the middle 95% of the distribution of $D_{bm}$ values that minimized these 1000 simulated sets as the estimated range. This error estimate, therefore, accounts for uncertainties in our own observations of biofilm growth and does not account for errors in other model parameters obtained from the literature. Holding this optimized value of $D_{bm}$ fixed, we performed a 2-D raster search across the parameter values for $p_{bla}$ (beta-lactamase production rate) and $k_{eq}$ (equilibration rate of ampicillin between biofilm and media) to minimize the discrepancy between simulated growth extents and those observed in experimental observations across all ampicillin concentrations, applying the same approach as for $D_{bm}$ (Supplementary Note 8 for more details on parameter search).

## Statistics and reproducibility

Biofilm imaging which was performed in triplicate (confocal) and quadruplicate (widefield) to enable replication statistics while maintaining a feasible scale for the total experiment. No statistical method was used to predetermine the sample size. No data were excluded from the analyses. The experiments were not randomized. The Investigators were not blinded to allocation during experiments and outcome assessment.

## Reporting summary

Further information on research design is available in the Nature Portfolio Reporting Summary linked to this article.

## Data availability

Confocal imaging data generated in this work are deposited to Bio-Image Archive under BioImages accession number S-BIAD1341. Newly generated plasmids and their associated sequences are available on Addgene [https://www.addgene.org/Ingmar_Riedel-Kruse]. Source data are provided with this paper.

## Code availability

Code used for analysis and visualization are shared on Github [https://github.com/xiaofanjin/biofilmCommunities][77].

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

## Acknowledgements

The authors thank D. Glass, H. Kim, E. Howley, S. Costan, P. Nutini, and A. Spormann for helpful suggestions, as well as K. Huang and K. Ng for confocal microscopy support. The authors furthermore acknowledge the support from Stanford Bio-X Bowes Fellowship (XJ) and NSERC PGS Fellowship (XJ), as well as the American Cancer Society RSG-14-177-01 (IHRK), NIH 1R01GM145893-01A1 (IHRK), and NSF 2214020 (IHRK).

## Author contributions

X.J. contributed to experiments, modeling, and data analysis. X.J. and I.H.R.-K. contributed to the design and implementation of the research, to the interpretation of the results, and to the writing of the manuscript.

## Competing interests

The authors declare no competing interests.
