## [Transparent Peer Review file · Nature Communications]

Optogenetic patterning generates multi-strain biofilms with spatially distributed antibiotic resistance

Corresponding Author: Professor Ingmar Riedel-Kruse

Version 0:

Reviewer comments:

Reviewer #1

(Remarks to the Author)

The manuscript by Jin and Riedel-Kruse investigates spatial organization of microbial biofilms using an updated biofilm lithography (MBL) toolkit that uses photo-masked illumination and optogenetics to engineer multi-strain communities with specific spatial architectures. In particular, they demonstrate the ability to create heterogeneous patterns using two approaches: successive adhesion (fig 1) and differentiation (fig 2) (which can also be used together). As proof of principle, they apply these techniques to two situations: 1) ecological competition and growth between patterned red and green communities in an initially intersecting configuration (figure 3), and 2) biofilms comprised of antibiotic sensitive and antibiotic resistant biofilms, where resistance is cooperative because it is driven by a diffusible enzyme synthesized in the resistant cells (figure 4). In both cases, they compare their results to the those from a biophysical reaction-diffusion-type model of microbial growth, which captures the main features of their experiment and allows them to estimate several interesting biophysical parameters.

Overall, this is an interesting paper that develops a powerful method that could be used to study many problems in quantitative microbial ecology. The paper is well written and easy to follow, and the method is an exciting development that will interest many researchers in microbiology and related fields. I have mostly minor suggestions for the authors to consider.

- As with all techniques, there are some limitations, and I think the authors might consider discussing them in more detail in the main text discussion. For example, how do the time-scales required to implement one of the successive patterning steps (i.e. the overnight light exposure) impact the measurements that can be done and/or their interpretation? Are there potential differences in physiological states of, say, the red vs the green cells that result from one being patterned first and staying on the surface for 24 hours longer? Figures 3e and 3f suggest it's not a big factor, at least in this example, but it might be worth briefly discussing.

- For the zone of protection estimates: the reported length scale (70 microns) will of course depend substantially on the size, number, and arrangement of resistant cells, biophysical properties of the biofilm (and diffusion of drug into it), the timescales you're considering (here set by the penetration rate of ampicillin into the biofilm) and other factors; I worry that the specific number, if given alone, could be easily misconstrued. I would recommend emphasizing that this estimate is specific to this experimental arrangement and consider not reporting it alone in the abstract, at least without some context. Of course, the point is that the reported experimental approach can potentially measure this length scale (and that for other ecological interactions) in any engineered setup...and that's great.

- Are the authors able to comment on the plausibility of the ~ 10 hr penetration constant for ampicillin in biofilms in light of other results in the literature? (e.g. doi: 10.1128/aac.44.7.1818-1824.2000, or <https://doi.org/10.1093/jac/dkq257>). Drug penetration has been studied (though by very different means) and I'm wondering if that previous data can be used to back out estimates you could compare to your results.

- Since there are multiple free parameters, it is possible that the procedure is globally optimizing in a way that gives non-physical (or at least inaccurate) estimates for the individual parameters. This would be possible, in principle, even if the model of the biophysical process were exact and perfectly correct (unless we know for sure it's a convex optimization with a unique optimum), which of course no model is. The authors might briefly discuss this limitation and urge some caution with the interpretation of the individual parameters estimated from these multi-variable fits.

- In the model for antibiotic action, the drug slows the birth rate (rather than, say, increasing the death rate). I think this is reasonable, especially given the nice agreement between model and data. But the authors might wish to discuss limitations of that assumption. For example, beta lactams lead to killing rates that are proportional to growth rates because of length-dependent lysis (<https://doi.org/10.1073/pnas.1719504115>, DOI:10.15252/msb.202211475) and therefore may not be fully consistent with the proposed model (which again, I think is ok; the model works well at the length and timescales being described). The goal of the current paper is not, of course, to provide a general and wholly realistic biophysical model, but it might be worth discussing the fact that 1) different “microscopic” models could lead to similar macroscopic behavior at this “tissue” scale and 2) even the most detailed models leave out a great deal of mechanistically known details. (These points offer another reason to use caution in interpreting specific parameter values in a strict biophysical sense, rather than as effective parameters).

Reviewer #2

(Remarks to the Author)

The manuscript of Jin & Riedel-Kruse introduces new possibility in the orthogonal patterning of multiple strains in biofilms. Two approaches using optogenetic and chemo-optogenetic circuits enable the sequential deposition and differentiation of different strains. Thereby, the study introduces interesting new concepts and possibilities in developing multi-species biofilms. The study demonstrates two cases in which competing and cooperating strains show spatially governed growth dynamics. The conclusions on the bacterial relationships here including the modeling need to be critically evaluated and a number of technical points need to be improved before publication.

- 1) Figure 1b-d: The authors should characterize the biofilm microstructure in greater detail for their two-strain biofilms. In particular, higher magnification images in the z-direction including also the GR region would provide better insight into the structure. Questions that remain unanswered include if the biofilm thickness and biomass are similar in the R, RG and G regions and what the ratio and spatial distribution of the GFP and RFP positive bacteria are.
- 2) It would be important to characterize if the colonizer and the successor strain are layered or intermixed in the GR regions in Figure 1, 3 and 4 using the optogenetic circuit and in the RB region in Figure 2 using the chemo-optogenetic circuit.
- 3) The strategy of differentiation in the chemo-optogenetic circuit highly depends on the expression with illumination and degradation kinetics of LuxR and the authors have added a degradation tag to accelerate the degradation of LuxR expressed during the first illumination step. It would be important to characterize the dynamics of LuxR activity and if the degradation tag indeed resulted in the desired effect. Moreover, there is noticeable mRuby (not RFP) expression in the B areas in Figure 2c (50% of RB). The statement that the expression is “exclusively” limited to the RB region should be moderated.
- 4) In Figure 3 the growth of the GFP and RFP bacteria is compared in different locations within the pattern assuming that they have initially the same density. This is clearly not the case looking at the images in Figure 3c (GFP signal is lower in the GR region compared to the G region) and the quantification in Figure 1c (the Hoechst signal is similar, i.e., the total bacterial density is similar in the GR and G region). These differences in initial density of the bacteria have to be taken into consideration in the interpretation of the data. Moreover, earlier the authors mention the negative impact that anaerobic zones have on the fluorescence signal. Therefore, it is problematic to directly use the fluorescence signal to measure growth and preferably other methods would be used verify some of the conclusions.
- 5) The statistical analysis of the data in Figures 3 and 4 is lacking and sufficient details should be added. For figure 3 it is written that $n=6$ but only three data points are visible per time point. It would also be important to specify if data is acquired from different regions in the same sample or if these represent biological replicates.
- 6) It is not clear how the authors come to the conclusion that “The high degree of concordance observed between experimental data and numerical simulation...” building on a parameter that is obtained from the simulation ($D_{bm} = 0.149 \mu\text{m}^2/\text{s}$) but was not determined experimentally.
- 7) The differences in the growth of ampicillin sensitive bacteria in Figure 4b and c show a high variance and statistical analysis needs to be provided to prove the strong conclusions drawn from these.
- 8) The most prominent concern is on the biophysical modeling because it presents with numerous discrepancies, which are already visible through the inconsistencies in the units of extracted parameters. For example, “beta lactamase production per bacterial dry mass ($\sim 3 \cdot 10^{-9}/\text{s}$)” mentioned in the abstract is later given as “ K_{prod} of $3.07 \cdot 10^{-9} \text{ gbla}/\text{gdrybiomass}/\text{s}$ ” (page 12, line 345) and “equilibration rate K_{eq} of $2.86 \cdot 10^{-5} /\text{s}$, corresponding to a time constant on the order of 10h.”, where it is unclear what the quantity in what unit is changing per second and what the meaning of “time constant” is. On this note, the authors should stick to the convention, where rates should be denoted as “ v ” with a unit of mass or concentration/time, rate constants as “ k ” with a unit of 1/time for first order kinetics and equilibrium constants “ K ”.
- 9) One of the major problems is in the estimation of the parameters for the beta lactamase (bla) production and the consideration of its kinetics. In reaction kinetics, one should use the concentrations of molecules and not their mass because concentrations of starting molecules, intermediates and products can be converted into each other considering the reaction stoichiometry, which is not true for mass. Moreover, the model includes a myriad of assumptions that are not explained and only listed in a table. The estimation of “the beta lactamase production rate normalized by dry cellular biomass” is very far fetched and as a critical parameter should be determined in the experimental context of the biofilm.

Minor points:

- 1) The terminology of colonizer and successor should be used consistently throughout the manuscript and not be mixed as “successor colonizer”.
- 2) The data for 300 $\mu\text{g}/\text{mL}$ and 3 mg/mL ampicillin is not shown in Figures 4c and 4e, respectively, but referred to in the text. The ampicillin range in the caption of Figure 4 is incorrect. It would be advisable to consistently use mg/mL throughout the manuscript.

3) Microscopy images similar to Figure 3c should also be provided for the growth of RFP bacteria at the boundary and the experiments in Figure 4.

4) The patterns of the three different strains in Figure 2d should be quantified in their contrasts as done for patterns of two strains.

Reviewer #3

(Remarks to the Author)

This is an important study that reports an exciting technique for biofilm studies. The study is well done and clearly reported. No flaws in data analysis, interpretation, or modeling were found.

Please see attached comments for additional comments and suggestions.

Version 1:

Reviewer comments:

Reviewer #1

(Remarks to the Author)

The revisions have improved and clarified several important points in the manuscript. I have no additional suggestions and congratulate the authors on a nice study.

Reviewer #2

(Remarks to the Author)

The authors have thoroughly addressed all the reviewers' comments. I recommend the publication of the current version.

Reviewer #3

(Remarks to the Author)

The authors have addressed all of my concerns in the revision. No additional changes are required.

We would like to thank the reviewers for their time and thoughtful comments. Please find below a copy of the original comments, alongside our responses (italicized / in blue) detailing the corresponding changes in the manuscript. We carried out the suggested changes, and we believe we were able to address all the comments in full.

REVIEWER COMMENTS

Reviewer #1 (Remarks to the Author):

The manuscript by Jin and Riedel-Kruse investigates spatial organization of microbial biofilms using an updated biofilm lithography (MBL) toolkit that uses photo-masked illumination and optogenetics to engineer multi-strain communities with specific spatial architectures. In particular, they demonstrate the ability to create heterogeneous patterns using two approaches: successive adhesion (fig 1) and differentiation (fig 2) (which can also be used together). As proof of principle, they apply these techniques to two situations: 1) ecological competition and growth between patterned red and green communities in an initially intersecting configuration (figure 3), and 2) biofilms comprised of antibiotic sensitive and antibiotic resistant biofilms, where resistance is cooperative because it is driven by a diffusible enzyme synthesized in the resistant cells (figure 4). In both cases, they compare their results to the those from a biophysical reaction-diffusion-type model of microbial growth, which captures the main features of their experiment and allows them to estimate several interesting biophysical parameters.

Overall, this is an interesting paper that develops a powerful method that could be used to study many problems in quantitative microbial ecology. The paper is well written and easy to follow, and the method is an exciting development that will interest many researchers in microbiology and related fields. I have mostly minor suggestions for the authors to consider.

- As with all techniques, there are some limitations, and I think the authors might consider discussing them in more detail in the main text discussion. For example, how do the time-scales required to implement one of the successive patterning steps (i.e. the overnight light exposure) impact the measurements that can be done and/or their interpretation? Are there potential differences in physiological states of, say, the red vs the green cells that result from one being patterned first and staying on the surface for 24 hours longer? Figures 3e and 3f suggest it's not a big factor, at least in this example, but it might be worth briefly discussing.

- *Thank you for this suggestion. We have expanded our discussion to highlight differences between colonizer and successor strains, in particular noting that “...in our current sequential patterning approach, there were additional physiological differences between the IC-RFP and SS-GFP strains in the final patterned biofilms that resulted from their different patterning order and protocols, e.g., IC-GFP bacteria were patterned on a naive surface and stayed on the surface one day longer compared to SS-GFP bacteria which were patterned on a pre-colonized surface. Additionally, patterning of SS-GFP occurred in the presence of TMP, with pDawn in SS-GFP additionally driving expression of DHFR. This increased the patterning efficiency of SS-GFP strain as growth (including that of planktonic cells) was minimized in unilluminated regions, funneling biomass to the illuminated regions -- this was reflected in the observation that single-strain SS-GFP biofilms (where SS-GFP was patterned onto naive polystyrene) exhibited much more rapid growth than their single-strain IC-RFP biofilms (Supplemental Section S5). In sequential co-cultured biofilms, this increase was likely offset by the fact that SS-GFP bacteria were patterned on a pre-colonized (rather than naive) surface, resulting in similar observed growth between IC-RFP and SS-GFP biofilms (Fig. 3e,f)”*

- For the zone of protection estimates: the reported length scale (70 microns) will of course depend substantially on the size, number, and arrangement of resistant cells, biophysical properties of the biofilm (and diffusion of drug into it), the timescales you're considering (here set by the penetration rate of ampicillin into the biofilm) and other factors; I worry that the specific number, if given alone, could be easily misconstrued. I would recommend emphasizing that this estimate is specific to this experimental arrangement and consider not reporting it alone in the abstract, at least without some context. Of course, the point is that the reported experimental approach can potentially measure this length scale (and that for other ecological interactions) in any engineered setup...and that's great.

- *Thank you for this suggestion. We have rewritten the manuscript to clearly emphasize that the zone of protection estimates are specific to our model system, for instance adding in our discussion section that “...derived values such as the 70 μ m zone of protection are also specific to our synthetic biofilm system and would vary in other systems depending for instance on the rate of antibiotic penetration”, as well as removing the specific number from the abstract*

- Are the authors able to comment on the plausibility of the ~ 10 hr penetration constant for ampicillin in biofilms in light of other results in the literature? (e.g. doi: 10.1128/aac.44.7.1818-1824.2000, or <https://doi.org/10.1093/jac/dkq257>). Drug

penetration has been studied (though by very different means) and I'm wondering if that previous data can be used to back out estimates you could compare to your results.

- *We thank the reviewer for pointing these references out. We performed some rough calculations based on temporal data from Anderl et al 2000, which roughly corresponded to 1-3hr time constants (within an order of magnitude of our estimate in a different species / context) and added these to the paper: "Previous work estimated the penetration of ampicillin and ciprofloxacin in Klebsiella pneumoniae biofilms using the membrane filter disk approach [cite{anderl2000role}], measuring 50% penetration at 40 minutes and 120 minutes for ampicillin and ciprofloxacin respectively, corresponding to penetration time constants of 0.96hr and 2.98hr respectively assuming an exponential model."*

- Since there are multiple free parameters, it is possible that the procedure is globally optimizing in a way that gives non-physical (or at least inaccurate) estimates for the individual parameters. This would be possible, in principle, even if the model of the biophysical process were exact and perfectly correct (unless we know for sure it's a convex optimization with a unique optimum), which of course no model is. The authors might briefly discuss this limitation and urge some caution with the interpretation of the individual parameters estimated from these multi-variable fits.

- *Thank you for pointing out this limitation on the optimization for beta lactamase production rate, effective biofilm diffusivity, and antibiotic equilibration timescales, we have updated the text of the manuscript to reflect this by adding "We re-emphasize that these quantitative estimates were derived using multi-variable parameter optimization of biophysical modeling to observed data rather than direct experimental measurement, and while they are biologically plausible, nevertheless should be interpreted with a dose of caution." in the discussion*

- In the model for antibiotic action, the drug slows the birth rate (rather than, say, increasing the death rate). I think this is reasonable, especially given the nice agreement between model and data. But the authors might wish to discuss limitations of that assumption. For example, beta lactams lead to killing rates that are proportional to growth rates because of length-dependent lysis

(<https://doi.org/10.1073/pnas.1719504115>, DOI:10.15252/msb.202211475) and therefore may not be fully consistent with the proposed model (which again, I think is ok; the model works well at the length and timescales being described). The goal of the current paper is not, of course, to provide a general and wholly realistic biophysical model, but it might be worth discussing the fact that 1) different "microscopic" models

could lead to similar macroscopic behavior at this “tissue” scale and 2) even the most detailed models leave out a great deal of mechanistically known details. (These points offer another reason to use caution in interpreting specific parameter values in a strict biophysical sense, rather than as effective parameters).

- *We thank the reviewer for this insightful comment, we have updated the text of the manuscript to highlight that the way our model treats antibiotic effect involves simplifications that abstract away some biophysical realism, adding “...finer details of mechanism that have been abstracted away in our current simplified model. For instance, while we have modeled beta-lactam antibiotic effects as a reduction in growth rate (thus preserving model simplicity with death rate modeled by constant first order decay), in reality the inhibitory effect of beta-lactams is known to be mediated via growth-rate dependent increase in death rate \cite{lee2018robust,kim2023mapping} -- a mechanism which would require additional terms in the model to capture” in the discussion*

Reviewer #2 (Remarks to the Author):

The manuscript of Jin & Riedel-Kruse introduces new possibility in the orthogonal patterning of multiple strains in biofilms. Two approaches using optogenetic and chemo-optogenetic circuits enable the sequential deposition and differentiation of different strains. Thereby, the study introduces interesting new concepts and possibilities in developing multi-species biofilms. The study demonstrates two cases in which competing and cooperating strains show spatially governed growth dynamics. The conclusions on the bacterial relationships here including the modeling need to be critically evaluated and a number of technical points need to be improved before publication.

1) Figure 1b-d: The authors should characterize the biofilm microstructure in greater detail for their two-strain biofilms. In particular, higher magnification images in the z-direction including also the GR region would provide better insight into the structure. Questions that remain unanswered include if the biofilm thickness and biomass are similar in the R, RG and G regions and what the ratio and spatial distribution of the GFP and RFP positive bacteria are.

- *We thank the reviewer for this comment, we have added to Fig 1d GFP, RFP and Hoescht staining profiles in the z-direction for the N, R, RG and G regions to clarify biomass, GFP and RFP spatial distribution. We also include orthogonal slice images of the GR region in Fig 1b to provide better insight into the structure. Overall, the lack of fluorescent protein maturation beyond the top ~50 micron*

layer of the biofilm made it challenging to fully quantify the spatial distribution the two strains of throughout the biomass which we acknowledge as a limitation of our method in the results: “This lack of GFP and RFP signal in anaerobic layers largely prevented the characterization of the degree of vertical mixing or layering between the two strains in the Z-dimension” as well as in the discussion pointing to oxygen-independent flavin based FPs as a potential direction for future work: “The use of oxygen-independent fluorescent reporters \cite{drepper2007reporter} in place of RFP and GFP could also better illuminate spatial structure in anaerobic biofilm regions, a limitation of our current design which prevented the characterization spatial strain distribution in the Z-dimension.”

2) It would be important to characterize if the colonizer and the successor strain are layered or intermixed in the GR regions in Figure 1, 3 and 4 using the optogenetic circuit and in the RB region in Figure 2 using the chemo-optogenetic circuit.

- *We agree with the reviewer that this would be beneficial to characterize, however as discussed in the previous point the lack of oxygen in the deeper biofilm regions makes this challenging by preventing FP maturation. We added to our discussion that we hope that future work - for instance using oxygen independent fluorescent reporters - would circumvent this limitation. We do highlight in supplemental section S2 the spatial distribution between RFP and RFP bacteria in terms of interdigitation and co-exclusion distances based on spatial cross-correlation analysis, but again due to the discussed limitations, this is only done in the XY dimension in the upper biofilm region where FPs are able to mature.*

3) The strategy of differentiation in the chemo-optogenetic circuit highly depends on the expression with illumination and degradation kinetics of LuxR and the authors have added a degradation tag to accelerate the degradation of LuxR expressed during the first illumination step. It would be important to characterize the dynamics of LuxR activity and if the degradation tag indeed resulted in the desired effect. Moreover, there is noticeable mRuby (not RFP) expression in the B areas in Figure 2c (50% of RB). The statement that the expression is “exclusively” limited to the RB region should be moderated.

- *We agree with the reviewer that given the leaky expression of mRuby (thank you for catching the error re: mRFP) in B areas, it is advisable to moderate the corresponding statement in the text, which now reads: “Within in this subregion, expression of luxR via pDawn in conjunction with presence of AHL was designed to drive up-regulation of the fluorescent reporter mRuby2 \cite{Lam2012} via the*

pLux promoter, which effectively encoded an AND gate requiring both optical stimulation (to generate LuxR) and AHL (Fig. 1ref{fig:differentiation}a). ”, and “Quantification of fluorescent signal indicated significant up-regulation of mRuby2 expression in (RB) region relative to (B) and (N) regions, though leaky expression was evident in the (B) region”. The decision to add an LVA tag was made early on in prototyping the design, after an initial design without the LVA tag failed to differentiate - with this initial design, fluorescent signal was present throughout the entire patterned biofilm, not just the secondary illuminated regions. This led to the hypothesis that luxR produced in the initial biofilm patterning illumination step remained present at high enough concentrations that it was not required for secondary illumination to drive fluorescent protein expression, thus motivating the addition of the LVA tag. We included a description below for interest:

Early characterization / tuning experiment of differentiation circuit, testing behavior with and without LVA tag on LuxR:

A couple notes: this early version of the circuit used GFP instead of mRuby2. Also this used an earlier version of the protocol without trimethoprim in the media during the differentiation step, which meant that we observed problems with overgrowth. Also at this point in the project we were still playing around with illumination patterns so the two experiments used different illumination patterns for the initial biofilm patterning (polka dots for -LVA, vertical stripes for +LVA, both used horizontal stripes for secondary/differentiation step). Despite all this, what was clear from widefield microscopy results was that addition of LVA tag improved circuit behavior so that biofilm regions illuminated during the secondary differentiation step were brighter (in terms of GFP) than biofilm regions not illuminated during the secondary differentiation step.

4) In Figure 3 the growth of the GFP and RFP bacteria is compared in different locations within the pattern assuming that they have initially the same density. This is clearly not the case looking at the images in Figure 3c (GFP signal is lower in the GR region compared to the G region) and the quantification in Figure 1c (the Hoechst signal is similar, i.e., the total bacterial density is similar in the GR and G region). These differences in initial density of the bacteria have to be taken into consideration in the interpretation of the data. Moreover, earlier the authors mention the negative impact that anaerobic zones have on the fluorescence signal. Therefore, it is problematic to directly use the fluorescence signal to measure growth and preferably other methods would be used verify some of the conclusions.

- *Due to concern of the points raised here, we did not directly use an increase in fluorescent signal or Hoechst as a proxy for biofilm growth. Instead, we observed that biofilms expanded over time in the X-Y dimension, and used the fluorescent signal to track this spatial expansion of the biofilm as a measurement of growth. We agree with the reviewer that the biophysical model as currently implemented is simplistic and does not account for density differences in the initial biofilm. We have updated the discussion to openly acknowledge this and present it as a potential direction for future work: “Future models can also better reflect subtle differences between constituent strains -- for instance, apart from antibiotic resistance, our current model simplistically considers IC-RFP and SS-GFP strains as identical. However, in our current sequential patterning approach, there were additional physiological differences between the IC-RFP and SS-GFP strains in the final patterned biofilms that resulted from their different patterning order and protocols, such as initial density differences between IC-RFP and SS-GFP biofilms across G, R, and GR regions that were not captured in the current simplified biophysical model.”*

5) The statistical analysis of the data in Figures 3 and 4 is lacking and sufficient details should be added. For figure 3 it is written that n=6 but only three data points are visible per time point. It would also be important to specify if data is acquired from different regions in the same sample or if these represent biological replicates.

- *We thank the reviewer for pointing out these deficiencies. We have clarified in the methods how replicates were obtained (different regions in the same sample), stating that “Biofilms were cultured in M63 culture medium supplemented with 500ng/ml Hoechst 34580 stain in 6 well plates, and imaged every 2h for a 36h timecourse using Zeiss LSM 880 Confocal Microscope with a 10x objective, in a heated 30°C microscopy chamber. Hoechst, GFP and RFP signals were captured using excitation / emission wavelengths of 405/455nm, 488/530nm and 561/626nm respectively. The imaging focused on a 850µm by 850µm region of interest where a vertical GFP+ biofilm stripe intersected a horizontal RFP+ biofilm stripe. Replicates were obtained by imaging 3 such regions in each well / condition.” We have also updated our statistical analysis and included this in Supplemental Section “Quantification and statistical comparison of RFP+and GFP+ biofilm expansion”. All code for the statistical analysis has been updated and uploaded to the github repo <https://github.com/xiaofanjin/biofilmCommunities>*

6) It is not clear how the authors come to the conclusion that “The high degree of concordance observed between experimental data and numerical simulation...” building on a parameter that is obtained from the simulation ($D_{bm} = 0.149 \mu\text{m}^2/\text{s}$) but was not determined experimentally.

- *We agree with the reviewer and we have moderated the language to say that “The ability of our biophysical model to capture behavior observed in experimental data suggests...”*

7) The differences in the growth of ampicillin sensitive bacteria in Figure 4b and c show a high variance and statistical analysis needs to be provided to prove the strong conclusions drawn from these.

- *Thank you for this comment, we have updated our statistical analysis and included this in Supplemental Section “Quantification and statistical comparison of RFP+ and GFP+ biofilm expansion”*

8) The most prominent concern is on the biophysical modeling because it presents with numerous discrepancies, which are already visible through the inconsistencies in the units of extracted parameters. For example, “beta lactamase production per bacterial dry mass ($\sim 3 \cdot 10^{-9}/\text{s}$)” mentioned in the abstract is later given as “ K_{prod} of $3.07 \cdot 10^{-9} \text{ gbla}/\text{gdrybiomass}/\text{s}$ ” (page 12, line 345) and “equilibration rate K_{eq} of $2.86 \cdot 10^{-5}/\text{s}$, corresponding to a time constant on the order of 10h.”, where it is unclear what the quantity in what unit is changing per second and what the meaning of “time constant” is. On this note, the authors should stick to the convention, where rates should be denoted as “ v ” with a unit of mass or concentration/time, rate constants as “ k ” with a unit of 1/time for first order kinetics and equilibrium constants “ K ”.

- *We thank the reviewer for this helpful feedback, we have updated the nomenclature and units for parameters to align with convention, including*
 - *Cell biomass 2D density $\rho \rightarrow b$ (dimension M/L^2)*
 - *Nutrient concentration $c_{\text{nut}} \rightarrow p_{\text{nut}}$ (dimension M/L^3 , switched to p to emphasize mass concentration)*
 - *Ampicillin concentration $c_{\text{amp}} \rightarrow \text{camp}$ (dimension $M/L^3 \rightarrow N/L^3$ – switched from mass to molar concentration)*
 - *Beta-lactamase concentrations $c_{\text{bla}}(\text{Media, Biofilm}) \rightarrow \text{cbla}(\text{Media, Biofilm})$ (dimension $M/L^3 \rightarrow N/L^3$ – switched from mass to molar concentration)*
 - *Ampicillin culture-media-to-biofilm equilibration rate constant $K_{\text{eq}} \rightarrow k_{\text{eq}}$ (dimension $1/T$)*

- Beta-lactamase production rate $k_{\text{prod} \rightarrow \text{pbla}}$ (dimension $M_{\text{bla}}/M_{\text{biomass}}/T \rightarrow$ molecule/cell/T – molecular production per cell selected as a more intuitive basis, model uses Avogadro's constant to convert from molecular basis to molar basis for downstream calculations)
- Catalysis rate constant for beta-lactamase mediated ampicillin degradation $k_{\text{catBla} \rightarrow \text{kcatBla}}$ (dimension $1/T$)
- Additionally, we have updated the Methods to clarify dimensions for each variable and parameter, for example“
 - b_G is the 2D biomass density of green bacteria representing SS-GFP biofilm producing beta-lactamase, resistant to ampicillin (dimension M/L^2)
 - b_R is the 2D biomass density of red green bacteria representing IC-RFP biofilm, susceptible to ampicillin (dimension M/L^2)
 - c_{blaMedia} is the local molar concentration of ampicillin in the culture medium (dimension N/L^3)
 - $c_{\text{blaBiofilm}}$ is the local molar concentration of ampicillin in the biofilm (dimension N/L^3)
 - c_{bla} is the local molar concentration of beta-lactamase enzyme in the biofilm (dimension N/L^3)
 - ρ_{nut} is the local mass concentration of nutrients (dimension M/L^3)”

9) One of the major problems is in the estimation of the parameters for the beta lactamase (bla) production and the consideration of its kinetics. In reaction kinetics, one should use the concentrations of molecules and not their mass because concentrations of starting molecules, intermediates and products can be converted into each other considering the reaction stoichiometry, which is not true for mass. Moreover, the model includes a myriad of assumptions that are not explained and only listed in a table. The estimation of “the beta lactamase production rate normalized by dry cellular biomass” is very far fetched and as a critical parameter should be determined in the experimental context of the biofilm.

- Thank you for noting these deficiencies. We have converted the beta-lactamase and ampicillin values in the model from a mass to molar basis. Nutrient and cell biomass remain as a mass basis, as nutrient to biomass yields are typically reported on a mass basis. We have updated the beta lactamase production per bacterial dry mass ($\sim 3 \cdot 10^{-9}/s$) to a more biologically meaningful beta lactamase production per cell (0.013 molecule / cell / second). We have also updated the Methods to point to the Supplemental Section where parameters and initial conditions for the numerical simulations are listed, “Using a forward time centered space numerical approach, we performed numerical simulation of this model, using biophysically realistic parameter values and initial conditions --

drawn from literature when available (Supplemental Section S7)”. Finally, we also updated the language in our discussion to note caution on the interpretation of fitted parameters such as beta lactamase production and highlight that it was obtained indirectly via numerical optimization/simulation rather than direct measurement, for instance “We re-emphasize that these quantitative estimates were derived using multi-variable parameter optimization of biophysical modeling to observed data rather than direct experimental measurement, and while they are biologically plausible, nevertheless should be interpreted with a dose of caution.”

Minor points:

1) The terminology of colonizer and successor should be used consistently throughout the manuscript and not be mixed as “successor colonizer”.

- *Thank you for pointing this out, we have named the three key strains in this manuscript as IC-RFP (initial colonizer strain with mRFP marker), SS-GFP (secondary successor strain with GFP marker), and DS-mRuby2 (differentiation strain with mRuby2 marker), and made the terminology consistent throughout the text.*

2) The data for 300 µg/mL and 3 mg/mL ampicillin is not shown in Figures 4c and 4e, respectively, but referred to in the text. The ampicillin range in the caption of Figure 4 is incorrect. It would be advisable to consistently use mg/mL throughout the manuscript.

- *Thank you for pointing this out, we have made the ampicillin units consistent as mg/mL, and updated the figure references in the text where 0.3 and 3mg/mL ampicillin are discussed.*

3) Microscopy images similar to Figure 3c should also be provided for the growth of RFP bacteria at the boundary and the experiments in Figure 4.

- *We have included corresponding IC-RFP images of Fig 3c in supplemental section S4. Additionally, we have uploaded all microscopy data to a BioImage Archive*

4) The patterns of the three different strains in Figure 2d should be quantified in their contrasts as done for patterns of two strains.

- *Thank you for this suggestion, this quantification has been performed and added to supplemental section S3*

Reviewer #3 (Remarks to the Author):

This is an important study that reports an exciting technique for biofilm studies. The study is well done and clearly reported. No flaws in data analysis, interpretation, or modeling were found.

Review of “Optogenetic patterning generates multi-strain biofilms with spatially distributed antibiotic resistance” by Jin and Riedel-Kruse. This manuscript reports a new synthetic biology tool to spatially pattern the placement and activity of multiple strains of bacteria. The technique uses a combination of optogenetic circuits to spatially regulate the expression of an adhesin protein and chemical inducers to temporally regulate expression of other genes. Combining these two strategies enables the creation of biofilms with distinct phenotypes in predefined regions of space. These patterned biofilms mimic the spatially heterogeneity of phenotypes in natural biofilms. The manuscript verified the ability to achieve such spatial patterning. This new tool is then applied towards understanding antibiotic resistance in mixed biofilms. Strains of antibiotic sensitive and resistant *E. coli* were patterned on a surface, and the ability of the resistant strain to impart partial resistance to the sensitive strains was demonstrated. A biophysical model was developed to probe and better understand these local growth effects. Overall, the study is well conceived and executed and demonstrates the utility of this new tool to quantitatively examine interactions within mixed biofilms. Many of the proof of concept and control experiments were impressive (such as the PBS control and the comparison of ampicillin and kanamycin resistant cells). The authors are the pioneers of optogenetic patterning of bacterial biofilms, and this latest study reports an important extension of this prior work that has a strong potential to impact biofilm research moving forward. Adding these constructs to the addgene repository is appreciated.

Below are a few minor comments that should be addressed to improve clarity of the manuscript. Comments:

1. There is not much information on the Hoechst staining procedure. Is there only one possible Hoechst stain? How was the stain imaged?

- *Thank you for pointing out this oversight, we have updated our Methods section to specify that we used Hoescht 34580, as part of the description of the imaging protocol: “ Biofilms were cultured in M63 culture medium supplemented with 500ng/ml Hoechst 34580 stain, and imaged every 2h for a 36h timecourse using Zeiss LSM 880 Confocal Microscope with a 10x objective, in a heated (30°C)*

microscopy chamber. Hoechst, GFP and RFP signals were captured using excitation / emission wavelengths of 405/455nm, 488/530nm and 561/626nm respectively.”

2. It was a little confusing that in Figure 2 the schematic in “a” explained the results in b and c, but did not show the GFP patterning step shown in d. The authors might consider adding the step to pattern the GFP strain (which is already shown in 1a already), but they should use their own judgement whether such a change improves the figure. Maybe it should be too difficult to fit all 3 steps in the same panel.

- *Thank you for pointing this suggestion. We have created a new supplemental section S3 that includes a full schematic used to generate the 3-step patterning process used to generate figure 2d, as well as quantification of fluorescence in each region.*

3. Are the vertical lines in Figure 3d defined? It seems likely this represents the position of the biofilm boundary, but I can't find where this is stated. This comment is also relevant to Figure S3.

- *We thank the reviewer for pointing out this omission, we have updated the caption in Fig 3 and Fig S4 to state that “...grey and green vertical bars indicate extent of the expanding biofilm front inferred from fits at 0h and 10h respectively”*

4. Line 307: Repeated “experiments” in the sentence.

- *Thank you for catching this typo, it has been corrected.*

5. On line 466 in methods, it may be helpful to specify which AHL molecule was used in these experiments. The compound was listed much earlier in the manuscript, but this seems like a useful detail for the methods section.

- *Thank you for this suggestion, we have updated the Methods section to specify AHL used was 3-oxohexanoyl-homoserine lactone.*

6. The concept of biofilm diffusivity is a little confusing. Does this represent biofilm spread due to cell division? Is it the result of cell motility? Is it obvious that the biofilm diffusivity would be the same for all strains under all antibiotic conditions? A little more explanation of what this parameter physically represents would be helpful.

- *Thank you for this comment, we have updated the text to highlight this limitation that biofilm diffusivity term is an empirical parameter used to describe observed expansion of the biofilm front. For instance, when introducing this term in the results, we now state “...though it should be noted that bacterial diffusivity D_{bm} in our case solely exists as an empirical parameter used to describe observed expansion of the biofilm front, rather than a parameter with a well-described mechanistic underpinning”. In this sense, the diffusivity term is a simplified abstraction of underlying processes that contribute to biofilm spread, which could be captured more faithfully in more computationally intensive models such as agent based modeling – we make this explicit in the Discussion, adding that “...for instance, agent-based models could be used to capture biofilm spread more faithfully than our effective biomass diffusivity approach, incorporating distinct terms to account for passive and active motility (e.g., shoving vs. swarming)”*

7. The use of GFP+ and RFP+ was a little confusing, especially in the SI. Does GFP+ simply indicate a strain expressing GFP, or is it a strain with a specific set of plasmids (pDawn_Ag43 + pSuccessor)? Is RFP+ a strain with pDawn + pDifferentiation or does it have pDawn + Bba_J04450? Particularly in the SI, GFP+ and RFP+ are used frequently, but it wasn't always clear what strains were being used. I noticed this particularly in SI section 5. If the labels GFP+ and RFP+ do not point to a specific strain, the authors should ensure that for each experiment the precise strain used is clear.

- *Thank you for pointing this out, we have named the three key strains in this manuscript as IC-RFP (initial colonizer strain with mRFP marker), SS-GFP (secondary successor strain with GFP marker), and DS-mRuby2 (differentiation strain with mRuby2 marker), and made the terminology consistent throughout the text.*

8. Figure S1b seems to have formatting issues, or at least I can't understand the scheme for labelling the bars.

- *Thank you for catching this error, Figure S1 has been updated with proper labeling*

9. Near Figure S3 is the text “400\$upmum”.

- *Thank you for catching this typo, it has been corrected*

10. Figures S3 and S4 may benefit from the dashed lines used in the main text to show the boundary of the overlapping region (ie showing the edge of the RFP patterned region in the GFP pictures and the edge of the GFP patterned region in the RFP pictures).

- *Thank you for this comment, we have updated Figs S4 and S5 with dashed lines as suggested*

11. In SI Section 5, any thoughts why the GFP+ monoculture expanded by 220 microns but the RFP+ monoculture only expanded by 55 microns? This seemed strange given that in the coculture, both strains had similar expansion of about 35 microns.

- *Thank you for this comment. We do not have a definitive answer to this question, our best idea so far is that SS-GFP monoculture biofilms contain more initial biomass than IC-RFP monoculture, or for that matter SS-GFP biofilms in IC-RFP. As a result of this increased biomass, we speculate that the outward growth is faster. We speculate that this increased original biomass is due to the presence of TMP, and light-regulated DHFR production in SS-GFP that prevents growth (not just biofilm attachment) everywhere except illuminated regions, meaning nutrients are more efficiently funneled into biomass in illuminated regions. We have expanded our discussion to highlight these nuances between IC-RFP and SS-GFP strains, in particular noting “Our current successive biofilm patterning protocol means that IC-GFP bacteria were patterned on a naive surface and stayed on the surface one day longer compared to SS-GFP bacteria which were patterned on a pre-colonized surface. Patterning of SS-GFP also occurred in the presence of TMP, with pDawn in SS-GFP additionally driving expression of DHFR. This increased the patterning efficiency of SS-GFP strain as growth (including that of planktonic cells) was minimized in unilluminated regions, funneling biomass to the illuminated regions -- this was reflected in greater Hoechst staining intensity in G compared to R regions of the biofilm, as well as the observation that single-strain SS-GFP biofilms (where SS-GFP was patterned onto naive polystyrene) exhibited much more rapid growth than their single-strain IC-RFP biofilms (Supplemental Section S6). In sequential co-cultured biofilms, this increase was likely offset by the fact that SS-GFP bacteria were patterned on a pre-colonized (rather than naive) surface, resulting in similar observed growth between IC-RFP and SS-GFP biofilms (Fig. 3e,f).”*

12. The text in Figure S5 c-f was too small.

- *Thank you for noting this, we have increased text size in these panels.*

13. There is some inconsistency about italicizing *E. coli* throughout the document. From my experience, many reviewers pay extra attention to this particular detail.

- *Thank you for catching this, we have made italicization consistent*

Review of “Optogenetic patterning generates multi-strain biofilms with spatially distributed antibiotic resistance” by Jin and Riedel-Kruse.

This manuscript reports a new synthetic biology tool to spatially pattern the placement and activity of multiple strains of bacteria. The technique uses a combination of optogenetic circuits to spatially regulate the expression of an adhesin protein and chemical inducers to temporally regulate expression of other genes. Combining these two strategies enables the creation of biofilms with distinct phenotypes in predefined regions of space. These patterned biofilms mimic the spatial heterogeneity of phenotypes in natural biofilms. The manuscript verified the ability to achieve such spatial patterning. This new tool is then applied towards understanding antibiotic resistance in mixed biofilms. Strains of antibiotic sensitive and resistant *E. coli* were patterned on a surface, and the ability of the resistant strain to impart partial resistance to the sensitive strains was demonstrated. A biophysical model was developed to probe and better understand these local growth effects. Overall, the study is well conceived and executed and demonstrates the utility of this new tool to quantitatively examine interactions within mixed biofilms. Many of the proof of concept and control experiments were impressive (such as the PBS control and the comparison of ampicillin and kanamycin resistant cells). The authors are the pioneers of optogenetic patterning of bacterial biofilms, and this latest study reports an important extension of this prior work that has a strong potential to impact biofilm research moving forward. Adding these constructs to the addgene repository is appreciated.

Below are a few minor comments that should be addressed to improve clarity of the manuscript.

Comments:

1. There is not much information on the Hoechst staining procedure. Is there only one possible Hoechst stain? How was the stain imaged?
2. It was a little confusing that in Figure 2 the schematic in “a” explained the results in b and c, but did not show the GFP patterning step shown in d. The authors might consider adding the step to pattern the GFP strain (which is already shown in 1a already), but they should use their own judgement whether such a change improves the figure. Maybe it should be too difficult to fit all 3 steps in the same panel.
3. Are the vertical lines in Figure 3d defined? It seems likely this represents the position of the biofilm boundary, but I can’t find where this is stated. This comment is also relevant to Figure S3.
4. Line 307: Repeated “experiments” in the sentence.
5. On line 466 in methods, it may be helpful to specify which AHL molecule was used in these experiments. The compound was listed much earlier in the manuscript, but this seems like a useful detail for the methods section.
6. The concept of biofilm diffusivity is a little confusing. Does this represent biofilm spread due to cell division? Is it the result of cell motility? Is it obvious that the biofilm diffusivity would be the same for all strains under all antibiotic conditions? A little more explanation of what this parameter physically represents would be helpful.
7. The use of GFP+ and RFP+ was a little confusing, especially in the SI. Does GFP+ simply indicate a strain expressing GFP, or is it a strain with a specific set of plasmids (pDawn_Ag43 + pSuccessor)? Is RFP+ a strain with pDawn + pDifferentiation or does it have pDawn + Bba_J04450? Particularly in the SI, GFP+

and RFP+ are used frequently, but it wasn't always clear what strains were being used. I noticed this particularly in SI section 5. If the labels GFP+ and RFP+ do not point to a specific strain, the authors should ensure that for each experiment the precise strain used is clear.

8. Figure S1b seems to have formatting issues, or at least I can't understand the scheme for labelling the bars.

9. Near Figure S3 is the text "400^{µm}".

10. Figures S3 and S4 may benefit from the dashed lines used in the main text to show the boundary of the overlapping region (ie showing the edge of the RFP patterned region in the GFP pictures and the edge of the GFP patterned region in the RFP pictures).

11. In SI Section 5, any thoughts why the GFP+ monoculture expanded by 220 microns but the RFP+ monoculture only expanded by 55 microns? This seemed strange given that in the coculture, both strains had similar expansion of about 35 microns.

12. The text in Figure S5 c-f was too small.

13. There is some inconsistency about italicizing *E. coli* throughout the document. From my experience, many reviewers pay extra attention to this particular detail.